

# Development of the global maize production model MATCRO-Maize version 1.0

Marin Nagata[1], Astrid Yusara[2], Tomomichi Kato[3*], Yuji Masutomi[4]

[1]Graduate School of Global Food Resources, Hokkaido University, Sapporo, Hokkaido 060-0809, Japan
[2]Graduate School of Agriculture, Hokkaido University, Sapporo, Hokkaido 060-0808, Japan
[3]Research Faculty of Agriculture, Hokkaido University, Sapporo, Hokkaido 060-8589, Japan
[4]Center for Climate Change Adaptation, National Institute for Environmental Studies, Tsukuba, Ibaraki 305-8506, Japan

*Correspondence to*: Tomomichi Kato (tkato@agr.hokudai.ac.jp)

**Abstract.** Process-based crop models combined with land surface models are useful tools for accurately quantifying the impacts of climate change on crops while considering the interactions between agricultural land and climate. We developed a new process-based crop model for maize, named MATCRO-Maize, by incorporating leaf-level photosynthesis of C4 plants and adjusting crop-specific parameters into the original MATCRO model, which is a process-based crop model initially developed for paddy rice combined with a land surface model. The model was validated at both a point scale and a global 15 scale through comparisons with observational values. The validation at the point scale was conducted at four globally distributed sites. It showed statistically significant correlation for three variables (leaf area index: correlation coefficient (COR) of 0.76 with a p value < 0.01; total aboveground biomass: COR of 0.89 with a p value < 0.001; final yield: COR of 0.34 with a p value < 0.01). For the global scale validation, the simulated yield was statistically compared with the FAOSTAT data at the country level and total global level. Although the absolute value of the simulated yield tended to be 20 overestimated, MATCRO-Maize could capture spatial variability, as indicated by a COR of 0.58 (p value < 0.01) for the 30-year average yield comparison of the top 20 maize-producing countries. In addition, the comparisons of the interannual variability derived from detrended deviation were statistically significant for the total global yield (COR of 0.54 with p value < 0.01) and for half of the top 20 countries (COR of 0.64-0.90 with p value < 0.001 for 6 countries; COR of 0.50-0.51 with p value < 0.01 for 2 countries; COR of 0.48-0.55 with p value < 0.05 for 2 countries), which are comparable with those of 25 other global crop models. One of the reasons for this overestimation could be related to the strong nitrogen fertilization effect observed in MATCRO-Maize. With experimental field data under more comprehensive conditions, improvements in the functions of nitrogen fertilizer in the model would be needed to simulate the maize yield more accurately.



## 1 Introduction

Maize (*Zea mays* L.) is one of the most important cereals not only because of its large production (FAO, 2022) but also because of its various roles in human food, feed, and industrial uses. Maize has high photosynthetic efficiency as a C4 plant. It contains phosphoenolpyruvate (PEP) carboxylase in mesophyll cells, which concentrates $CO_2$ in bundle sheath cells. The concentrated $CO_2$ increases the relative amount of carboxylation versus oxygenation performed by ribulose-1,5-bisphosphate carboxylase/oxygenase (Rubisco) (Kanai and Edwards, 1999), allowing C4 plants to operate at lower stomatal conductance

rates than C3 plants (Sage, 1999). This mechanism results in high efficiencies of light, water, and nitrogen use (Knapp and Medina, 1999; Long, 1999). These features, such as multipurpose crops and high photosynthetic efficiency, enable the cultivated area to range over wide environments from wet to dry and from low to midlatitude. However, climate change impacts and climate-related extremes negatively affect the productivity of the agricultural sector, which leads to negative consequences for food security (Intergovernmental Panel on Climate Change (IPCC), 2023). Therefore, it is important to

accurately quantify the impact of climate change on crop growth and yield and to identify effective adaptation strategies to mitigate climate risk.

Process-based crop models are useful tools for climate change studies because they consider the response of the physiological processes of crop growth and development to the environment and management (Tubiello and Ewert, 2002). The ensemble of process-based crop model simulations has shown good agreement with observed maize yields both at the

site scale and at the global scale (Bassu et al., 2014; Jägermeyr et al., 2021), showing its potential to quantify the uncertainty in studies on the impacts of climate change on crop yields (Asseng et al., 2013). Crop models combined with land surface models (LSMs) or earth system models (ESMs) (as classified by Peng et al., 2017) have the ability to consider the effects of agricultural land on the climate globally through the exchange of fluxes of heat, water, and gases, as well as the effects of climate on crops. Some studies have revealed that agricultural land affects the climate through fluxes (Bondeau et al., 2007;

Levis et al., 2012; Maruyama and Kuwagata, 2010; Tsvetsinskaya et al., 2001) and subsequently affects crop production (Osborne et al., 2009). This indicates the importance of considering the interaction between agricultural land and climate to accurately quantify the impacts of climate change on crops. Despite this importance, few LSM/ESM-based crop models exist (Lin et al., 2021; Lombardozzi et al., 2020; Osborne et al., 2015; Wu et al., 2016).

MATCRO is a process-based crop growth model developed for C3 plants (Masutomi et al., 2016a, b; Yusara et al., in

prep). It was initially combined with a land surface model of Minimal Advanced Treatments of Surface Interaction and Runoff, called MATSIRO (Takata et al., 2003). MATSIRO is embedded in an earth system model, which is the Model for Interdisciplinary Research on Climate, Earth System version 2 for Long-term simulations called MIROC-ES2L (Hajima et al., 2020). MATCRO simulates crop growth based on leaf-level photosynthesis and parameterized crop-specific parameters determined from experimental data, and can run simulations both at a point scale and at a global scale. The model was

applied to assess the impact of climate change at the country and local levels (Kinose et al., 2020; Kinose and Masutomi, 2019) and was used in the study investigating factors to improve the simulation performance of global gridded crop models





(GGCMs) (Iizumi et al., 2021). MATCRO is applicable to other crops, including maize as a C4 plant, with adjusted parameters from experimental datasets and the literature.

We extended MATCRO for global maize yield simulation, called MATCRO-Maize, by adjusting crop-specific
parameters for maize and incorporating the C4 photosynthetic mechanism. MATCRO-Rice can simulate latent heat flux, sensible heat flux, net carbon uptake by crops, and rice yield, indicating its application in studies on climate change impacts as an LSM-based model (Masutomi et al. 2016b). However, this study focused only on crop growth and yields, omitting water and heat fluxes to increase computational efficiency. This paper aims to describe MATCRO-Maize in detail (Section 2) and model validation on simulated yields both at a point scale and at a global scale (Section 3), with a discussion of the
validation and model limitations (Section 4).

## 2. Model description

MATCRO consists of four modules: radiation, net carbon assimilation, crop growth, and soil water balance. It requires the following input data: (i) phenological data (i.e., crop calendar), (ii) water management data (i.e., the land is rainfed or irrigated), (iii) nitrogen fertilizer application data ($N_{fert}$) [kg N ha$^{-1}$], (iv) soil classification data (i.e., soil texture
classification), (v) annual $CO_2$ data [ppm], and (vi) 6 types of daily meteorological data: air pressure ($P_s$) [Pa], precipitation ($P_{rc}$) [kg m$^{-2}$ s$^{-1}$], specific humidity [$S_h$] [kg kg$^{-1}$], downwards shortwave radiation ($R_s$) [W m$^{-2}$], maximum, minimum, and mean air temperature ($T_{max}, T_{min}, T_a$) [K], and wind speed ($U$) [m s$^{-1}$]. Based on input data, MATCRO simulates crop growth during a growing period. It is controlled by the crop developmental stage ($D_{vs}$) based on (Bouman et al., 2001), which is the index used to quantify crop development. The final crop yield is determined by the dry weight of the storage
organ with a parameter ($K_{yld}$) when $D_{vs} = 1$. To adapt MATCRO for maize, crop-specific parameters and equations were improved, as shown in Table 1 and Eq. (1)–(35). The details are described in the following sections.

### 2.1 Photosynthetic mechanism

MATCRO calculates net carbon assimilation for the entire canopy ($A_n$) via the big-leaf model (Dai et al., 2004), where C3 leaf-level photosynthesis is separately calculated for sunlit and shaded leaves from the coupled photosynthesis–stomatal
conductance model (Collatz et al., 1991).

$A_n$ for the entire canopy is given by:

$$A_n = \overline{A}_{n,sn} L_{sn} + \overline{A}_{n,sh} L_{sh}, \tag{1}$$

where $\overline{A}_{n,sn}$ and $\overline{A}_{n,sh}$ represent the net carbon assimilation per unit leaf area [$\mu$ mol m$^{-2}$ s$^{-1}$] and where $L_{sn}$ and $L_{sh}$ represent the leaf area index (LAI) [m$^2$ (leaf) m$^{-2}$]. $sn$ and $sh$ indicate sunlit and shaded leaves, respectively. $\overline{A}_{n,sn}$ and $\overline{A}_{n,sh}$ are
defined in the following equations:



$$\overline{A}_{n,x} = \overline{A}_{g,x} - \overline{R}_{d,x}, \tag{2}$$

where $\overline{A}_{g,x}$ and $\overline{R}_{d,x}$ represent gross carbon assimilation and dark respiration per unit leaf area [$\mu$ mol m$^{-2}$ s$^{-1}$], respectively. Suffix $x$ means $sn$ or $sh$. $L_{sn}$ and $L_{sh}$ are determined following the approach of Masutomi et al., (2016a). $\overline{R}_{d,x}$ is calculated via the following equation (Bonan et al., 2011):

$$\overline{R}_{d,x} = 0.025 \, V_{cmax,x} \left( \frac{2^{(Tv-298.15)/10}}{1+\exp(1.3(Tv-328.15))} \right), \tag{3}$$

where $V_{cmax,x}$ [$\mu$ mol m$^{-2}$ s$^{-1}$] is the maximum rate of carboxylation and where $T_v$ is the leaf temperature [K] (assumed to be the same as the air temperature: $T_a$). $\overline{A}_{g,x}$ is determined by the smaller root of the following equations:

$$\beta_{cj}\overline{A}_{i,x}^2 - \left(\overline{A}_{c,x} + \overline{A}_{j,x}\right)\overline{A}_{i,x} + \overline{A}_{c,x}\overline{A}_{j,x} = 0, \tag{4}$$

$$\beta_{ip}\overline{A}_{g,x}^2 - \left(\overline{A}_{i,x} + \overline{A}_{p,x}\right)\overline{A}_{g,x} + \overline{A}_{i,x}\overline{A}_{p,x} = 0, \tag{5}$$

where $\beta_{cj}$ and $\beta_{ip}$ are the transition factors (Table 1) and where $\overline{A}_{i,x}$ [$\mu$ mol m$^{-2}$ s$^{-1}$] is the colimited photosynthesis. Here, we introduced the C4 leaf-level photosynthesis model based on Collatz et al., (1992) into MATCRO, in which some parameters were taken from Oleson et al., (2013) and Lawrence et al., (2020) (Table 1). In C4 photosynthesis, $\overline{A}_{c,x}$, $\overline{A}_{j,x}$, and $\overline{A}_{p,x}$ [$\mu$ mol m$^{-2}$ s$^{-1}$] represent Rubisco-limited, RUBP-limited, and PEP-limited photosynthesis, respectively, and are given by the following equations:

$$\overline{A}_{c,x} = V_{cmax,x} \,, \tag{6}$$

$$\overline{A}_{j,x} = \alpha(4.6Q_{ab,x}), \tag{7}$$

$$\overline{A}_{p,x} = k_{p,x}C_{i,x} \,, \tag{8}$$

where $C_{i,x}$ [ppm] is the internal leaf CO$_2$ concentration, $Q_{ab,x}$ [W m$^{-2}$] is the absorbed photosynthetically active radiation (PAR), $\alpha$ [mol mol$^{-1}$] is the quantum efficiency, and $k_{p,x}$ [$\mu$ mol m$^{-2}$ s$^{-1}$] is the initial slope of the CO$_2$ response curve for the C4 CO$_2$ response curve. $Q_{ab,x}$ is calculated from $R_s$ via the same methods as in Masutomi et al. (2016a) and is converted to photosynthetic photon flux by multiplying by 4.6 [$\mu$ mol (photons) J$^{-1}$]. $V_{cmax,x}$ and $k_{p,x}$ are functions of $T_v$ and are based on Lawrence et al. (2020),

$$V_{cmax,x} = f_v \, V_{cmax25,x} \left[ \frac{Q_{10}^{(T_v-298.15)/10}}{f_H(T_v)f_L(T_v)} \right], \tag{9}$$

$$f_H(T_v) = 1 + exp[S_1(T_v - S_2)], \tag{10}$$

$$f_L(T_v) = 1 + exp[S_3(S_4 - T_v)], \tag{11}$$

$$k_{p,x} = \begin{cases} k_{p25,x}Q_{10}^{(T_v-298.15)/10}, & V_{cmax25,x} > 0, \\ 0.7\mu, & V_{cmax25,x} = 0, \end{cases} \tag{12}$$

$$k_{p25,x} = 20000V_{cmax25,x} \,, \tag{13}$$

with $Q_{10} = 2$, $S_1 = 0.3 \, K^{-1}$, $S_2 = 313.15K$, $S_3 = 0.2 \, K^{-1}$, and $S_4 = 288.15K$ (Table 1). Notably, $k_{p,x}$ is adjusted to be $0.7\mu$ (Collatz et al., 1992) when $V_{cmax25,x} = 0$ because of the process of the photosynthesis calculation (Eq. (20)). $V_{cmax25,x}$



is the maximum Rubisco carboxylation rate per unit leaf area at 25℃ (the details are described in Section 2.2.2). $f_v$ is the water stress factor calculated in the soil water balance module, which indirectly affects $A_n$ through $V_{cmax,x}$ (Sellers et al., 1996). $f_v$ is derived from the following equations:

$$f_v = \sum_{i=1}^{NSL} \begin{cases} 1 * ETF(i), & FAW(i) > 0.45, \\ \frac{FAW(i)}{0.45} * ETF(i), & otherwise, \end{cases} \tag{14}$$


$$FAW(i) = \min\left(\frac{\max((WSL(i)-WILT),0)}{FC-WILT}, 1\right), \tag{15}$$

$$ETF(i) = \frac{3}{2}\frac{(z_{rt}^2-z^2)}{z_{rt}^3}, \tag{16}$$

where $NSL$ represents the number of soil layers, $ETF$ represents the fraction of transpiration from root distribution, $FAW$ represents the fraction of available water, $WSL$ represents the soil water content [m$^3$ m$^{-3}$], $WILT$ represents the wilting point, $FC$ represents the field capacity, and $z_{rt}$ and $z$ represent the root depth and the soil depth, respectively, for each layer.

MATCRO assumes $NSL = 5$, where each of the soil layers has thicknesses of 0.05, 0.2, 0.75, 1, and 2 [m], respectively. MATCRO uses the soil texture data as input data, where the soil is classified into 13 types, leading to differences in $WILT$ and $FC$ based on Campbell and Norman (1998). $WSL$ is calculated considering transpiration from the canopy, evaporation from the soil, and water flux (those calculations are the same as those of the original MATCRO). The $ETF$ calculation assumes that the root has no spatial orientation and is equally distributed in the soil (Masutomi et al., 2016a). $z_{rt}$ is

determined by the same calculation as the original MATCRO, where the crop-specific parameter ($z_{rt,mx}$) was changed to maize (Table 1). The conditional branch ($FAW(i) > 0.45$) is based on the FAO 56 guidelines (Allen et al., 1998).

Stomatal conductance influences $CO_2$ uptake during photosynthesis. MATCRO-Maize represents stomatal conductance for $CO_2$, $G_{sc,x}$ [$\mu$ mol m$^{-2}$ s$^{-1}$], based on Ball et al. (1987) as follows:

$$G_{sc,x} = \begin{cases} G_{0c} + G_{1c}R_h \frac{\overline{A_{n,x}}}{C_{s,x}}, & \overline{A}_{n,x} \geq 0, \\ G_{0c}, & otherwise, \end{cases} \tag{17}$$

where $C_{s,x}$ [ppm] is the $CO_2$ concentration at the leaf surface and where $R_h$ [-] is the relative humidity at the leaf surface. $G_{0c}$ and $G_{1c}$ are derived from parameters $b$ and $m$ (shown in Table 1), respectively, by adjusting their ratio of 1:1.6, which is the ratio of diffusivity of $H_2O$ to $CO_2$. Here, the leaf-level net carbon assimilation rate ($\overline{A}_{n,x}$), stomatal conductance for $CO_2$ ($G_{sc,x}$), and boundary layer conductance for $CO_2$ ($G_{bc}$) meet the following physical flux equations:

$$\overline{A}_{n,x} = G_{sc,x}(C_{s,x} - C_{i,x}), \tag{18}$$

$$\overline{A}_{n,x} = G_{bc}(C_a - C_{s,x}), \tag{19}$$

where $C_a$ [ppm] is the atmospheric $CO_2$ concentration. $G_{bc}$ is a function of air pressure ($P_s$ [$Pa$]) and the wind speed in the canopy ($U$ [m s$^{-1}$]) (Masutomi, 2023).

Here, $T_v$, $Q_{ab,x}$, $R_h$, $U$, and $C_a$ are environmental variables derived from input meteorological climate data. There are four relationships (Eqs. (2), (17)-(19)) in terms of internal variables ($\overline{A}_{n,x}$, $G_{sc,x}$, $C_{s,x}$, $C_{i,x}$). MATCRO for C3 photosynthesis





obtains analytical solutions from relationships via the method shown in Masutomi (2023). For C4 photosynthesis, it is also

possible to solve these equations analytically. In the case of Rubisco-limited and RuBP-limited photosynthesis, exact

expressions for $\overline{A}_{c,x}$ and $\overline{A}_{j,x}$ are obtained. Under $\overline{A}_{n,x} \geq 0$, PEP-limited photosynthesis ($\overline{A}_{p,x}$) can be represented by

quadratic equations by the algebraic procedures as follows:

$$0 = \{G_{bc}^2 G_{1c}R_h - G_{bc}G_{0c} - k_{p,x}(G_{0c} - G_{bc}G_{1c}R_h + G_{bc})\}\overline{A}_{p,x}^{\;2} + \{C_a G_{bc}^2 G_{0c} - G_{bc}G_{0c}R_d + G_{bc}^2 G_{1c}R_h R_d -$$

$$k_{p,x}C_a(G_{bc}^2 G_{1c}R_h - 2G_{bc}G_{0c} - G_{bc}^2)\}\overline{A}_{p,x} + C_a G_{bc}^2 G_{0c}(R_d - k_{p,x}C_a). \tag{20}$$

Under $\overline{A}_{n,x} < 0$, the PEP-limited photosynthesis rate can be expressed as

$$\overline{A}_{p,x} = \frac{k_{p,x}C_a - R_d}{1 + k_{p,x}\left(\frac{1}{G_{bc}} + \frac{1}{G_{0c}}\right)}. \tag{21}$$

According to these equations, in the case of PEP-limited photosynthesis, there are three possible solutions. Following the

criteria described by Masutomi (2023), only one analytical solution can be selected when the following requirements are

satisfied: (i) under $\overline{A}_{n,x} \geq 0$, the solution must be a positive or zero real solution, and under $\overline{A}_{n,x} < 0$, it must be a negative

real solution; (ii) $G_{sc,x} > 0$; and (iii) $C_i > 0$.

## 2.2 Crop-specific parameterization

### 2.2.1 Phenology

The crop growing period in MATCRO is controlled by $D_{vs}$ based on Bouman et al. (2001). Here, $D_{vs} = 0$ means sowing,

and $D_{vs} = 1$ means maturity (harvesting). It is calculated from the following equations:

$$D_{vs} = G_{dd}/G_{dd,m}, \tag{22}$$

$$G_{dd} = \int_0^t D_{vr}\,dt', \tag{23}$$

$$D_{vr} = \begin{cases} 0, & T_t < T_b \mid T_h \leq T_t, \\ T_t - T_b, & T_b \leq T_t < T_o, \\ \frac{(T_b - T_o)(T_t - T_h)}{(T_h - T_o)}, & T_o \leq T_t < T_h, \end{cases} \tag{24}$$

where $G_{dd}$ is the growing degree days at $t$ (time), $G_{dd,m}$ is the growing degree day at maturity, $D_{vr}$ is the developmental rate

at time $t$, and $T_t$ is the temperature at time $t$. $T_b$, $T_h$, and $T_o$ are the crop-specific cardinal temperatures (minimum,

maximum, and optimal temperatures for development, respectively, as shown in Table 1). $G_{dd,m}$ were parameterized for

each point scale simulation and global scale simulation (Section 2.3). In addition, one parameter that represents the timing of

flowering ($D_{vs,flw}$) was parameterized based on observational data for the point scale simulation (Table 1).

### 2.2.2 Leaf nitrogen and Rubisco capacity

**Maximum Rubisco carboxylation rate**



$V_{cmax25,x}$ used in the photosynthesis module (Section 2.1) is obtained by dividing the maximum Rubisco carboxylation rate at the canopy level ($V_{cmax25,x}(l)$) by $L_x$ separately for sunlit and shaded leaves based on Bonan et al. (2011). The vertical distribution of $V_{cmax25}(l)$, which is the sum of $V_{cmax25,sn}(l)$ and $V_{cmax25,sh}(l)$, follows the exponential profile:

$$V_{cmax25}(l) = V_{cmax25}(0)\exp(-K_n l), \tag{25}$$

where $V_{cmax25}(0)$ is the maximum Rubisco carboxylation rate at the canopy top, $K_n$ is a parameter for the vertical distribution of nitrogen (Table 1), and $l$ represents the LAI depth from the top. The maximum Rubisco carboxylation rate in sunlit leaves ($V_{cmax25,sn}(l)$) is also calculated by the same relationship considering the light distribution:

$$V_{cmax25,sn}(l) = V_{cmax25}(0)[1 - exp(-l(K_n + K))]\frac{1}{K_n+K}, \tag{26}$$

where $K$ is the direct beam extinction coefficient (the calculation is the same as that for Masutomi et al., 2016a).
$V_{cmax25,sh}(l)$ is given by the subtraction of Eq. (25) and Eq. (26).

Here, while Bonan et al. (2011) uses the fixed $V_{cmax25}(0)$ value in each plant functional type, $V_{cmax25}(0)$ in MATCRO is calculated dynamically as a function of specific leaf nitrogen ($S_{ln}$ [g N m$^{-2}$]). The function is established based on the experimental literature data. Notably, we applied the relationship between $S_{ln}$ and light-saturated $CO_2$ assimilation ($A_{max}$) from the literature, although MATCRO-Rice and MATCRO-Soybean utilize the direct relationship between $S_{ln}$ and
$V_{cmax25}(0)$ based on the experimental literature data. The reasons are that we assume that $A_{max}$ could be used as Rubisco-limited photosynthesis in C4 photosynthesis and that Rubisco-limited photosynthesis could be equal to the maximum Rubisco carboxylation rate from Eq. (6). Several studies have shown that $A_{max}$ has a close relationship with $S_{ln}$, as shown by the logistic equation for maize (Drouet and Bonhomme, 2004; Muchow and Sinclair, 1994; Paponov and Engels, 2003; Paponov et al., 2005; Sinclair and Horie, 1989; Vos et al., 2005). We used two functions from them for different $D_{vs}$ as
follows:

$$V_{cmax25}(0) = \begin{cases} 45.1 * \left\{ \frac{2}{1+\exp[-2.9*(S_{ln}-0.25)]} - 1 \right\}, & D_{vs} < D_{vs,flw}, \\ 40.2 * \left\{ \frac{2}{1+\exp[-1.41*(S_{ln}-0.43)]} - 1 \right\}, & D_{vs} \geq D_{vs,flw}, \end{cases} \tag{27}$$

where $D_{vs} < D_{vs,flw}$ represents the vegetative stage at which the equation was based on Vos et al. (2005); then, for the reproductive stage, the equation was from Drouet and Bonhomme (2004).

**Specific leaf nitrogen**

$S_{ln}$, which is used in the calculation of $V_{cmax25}(0)$, is obtained from the function of $D_{vs}$ in MATCRO. The function is established based on the observational data. We utilized the study by Muchow (1988), in which $S_{ln}$ was measured under various levels of $N_{fert}$ (0, 60, 120, 240, 420 [kg ha$^{-1}$]), as follows: (i) we traced $S_{ln}$ data using digitizer software (https://apps.automeris.io/wpd4/) and obtained the measurement and phenological data from the paper; and (ii) we conducted
the fitting based on the assumption that $S_{ln}$ linearly increased until flowering and then decreased towards maturity. The parameterization given by Eqs. (28)-(30) is shown in Figure 1.



$$S_{ln} = \begin{cases} \frac{S_{ln,mx} - S_{ln,plt}}{D_{vs,flw}} D_{vs} + S_{ln,plt}, \ D_{vs} < D_{vs,flw}, \\ \frac{S_{ln,matu} - S_{ln,mx}}{1 - D_{vs,flw}} (D_{vs} - 1) + S_{ln,matu}, \ D_{vs} \geq D_{vs,flw}, \end{cases} \tag{28}$$

where $S_{ln,matu}$, $S_{ln,mx}$, and $S_{ln,plt}$ are $S_{ln}$ at maturity, maximum $S_{ln}$, and $S_{ln}$ at planting, respectively (Table 1). $S_{ln,mx}$ and $S_{ln,matu}$ are empirically parameterized as functions of $N_{fert}$ as follows:

$$S_{ln,mx} = \begin{cases} -0.00001 \ N_{fert}^2 + \ 0.0064 \ N_{fert} + 0.6891, \ N_{fert} \leq 240, \\ 1.75, \ N_{fert} > 240. \end{cases} \tag{29}$$

$$S_{ln,matu} = \begin{cases} 0.001 \ N_{fert} + 0.57, \ N_{fert} \leq 240, \\ 1, \ N_{fert} > 240. \end{cases} \tag{30}$$

We set fixed values of 1.75 for $S_{ln,mx}$ and 1.0 for $S_{ln,matu}$ when $N_{fert}$ exceeds 240 [kg ha$^{-1}$], as $S_{ln,mx}$ and $S_{ln,matu}$ exhibit minimal increases beyond this threshold.

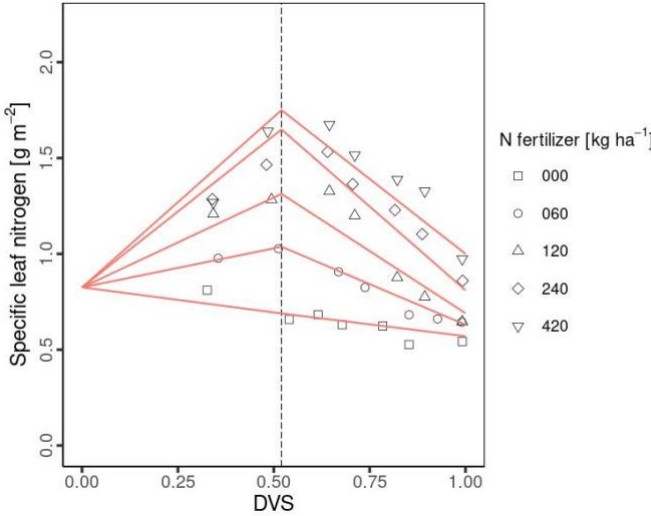

**Figure 1.** Relationship between developmental stage ($\boldsymbol{D_{vs}}$) and specific leaf nitrogen ($\boldsymbol{S_{ln}}$) in MATCRO-Maize. Shapes show observational data from Muchow (1988) with the 5 types of $\boldsymbol{N_{fert}}$: 0 kg ha$^{-1}$ (square), 60 kg ha$^{-1}$ (cycle), 120 kg ha$^{-1}$ (triangle), 240 kg ha$^{-1}$ (diamond), and 420 kg ha$^{-1}$ (inverted triangle). The red lines represent the fitted line parameters used in MATCRO-Maize, while the dashed line represents $\boldsymbol{D_{vs}}$ at flowering ($\boldsymbol{D_{flw}}$).

### 2.2.3 Crop growth

**Glucose partitioning**

MATCRO calculates crop growth by partitioning net carbon assimilation ($A_n$) in the form of glucose, which is calculated in the photosynthesis module (Section 2.1). Partitioned glucose is supplied through photosynthesis in leaves and remobilization from the stem. The ratio of glucose partition to each organ (leaf, stem, root, and storage organ; ear) depends on $D_{vs}$. The term "ear" in maize represents the reproductive organ responsible for grain development. The dry matter for each organ is obtained from the partitioned glucose considering the carbon fraction for each organ ($C_{glu,ear}$, $C_{glu,leaf}$, $C_{glu,rot}$, $C_{glu,stm}$ in



Table 1). We parameterized the partitioning ratio to leaf, stem, and ear based on the observational biomass data from Ciampitti et al. (2013a, b), whereas the ratio to shoots/roots was derived from the value from Penning de Vries et al. (1989). Figure 2 shows the partition ratio to the leaf ($P_{r,lef}$) and ear ($P_{r,ear}$) established via the following equations:

$$P_{r,lef} = \begin{cases} P_{lef}, & D_{vs} < D_{vs,lef1}, \\ \frac{P_{lef}(D_{vs,lef2}-D_{vs})}{D_{vs,lef2}-D_{vs,lef1}}, & D_{vs} < D_{vs,lef2}, \\ 0, & otherwise, \end{cases} \tag{31}$$


$$P_{r,ear} = \begin{cases} 0, & D_{vs} < D_{vs,ear1}, \\ \frac{D_{vs}-D_{vs,ear1}}{D_{vs,ear2}-D_{vs,ear1}}, & D_{vs} < D_{vs,lef2}, \\ 1, & otherwise, \end{cases} \tag{32}$$

where $D_{vs,lef1}$, $D_{vs,lef2}$, $D_{vs,ear1}$ and $D_{vs,ear2}$ represent the $D_{vs}$ at which the corresponding partition changes, as determined in Table 1 based on Figure 2, and where $P_{lef}$ is the ratio of glucose partitioned to glucose to the leaf from glucose partitioned to the shoot.

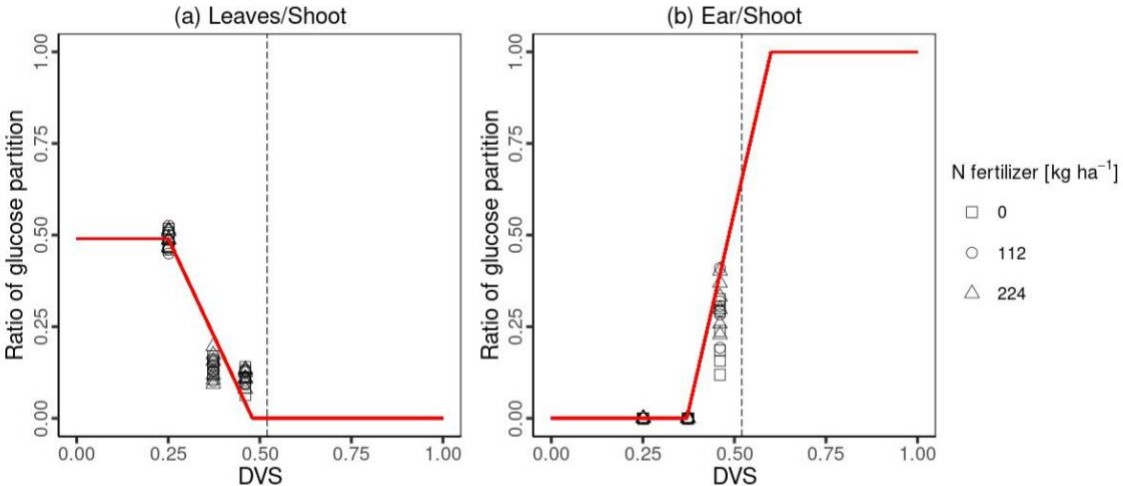

**Figure 2.** The ratio of glucose partitioning to leaves (a) and ears (b). Points show the ratio of glucose partition with different $N_{fert}$: 0 kg ha⁻¹ (square), 112 kg ha⁻¹ (cycle), and 224 kg ha⁻¹ (triangle) measured in Ciampitti et al. (2013a, b). The red lines in Figure 2 show the fitted line parameters used in MATCRO-Maize, while the dashed line represents $D_{vs}$ at flowering ($D_{vs,flw}$).

**Specific leaf weight**

The specific leaf weight ($S_{lw}$) is used to calculate the total leaf area index ($L$) in MATCRO. It is a function of $D_{vs}$ and is given by:

$$S_{lw} = S_{lw,mx} + \left(S_{lw,mn} - S_{lw,mx}\right)\exp\left(-k_{Slw}D_{vs}\right) \tag{33}$$

where $S_{lw,mn}$, $S_{lw,mx}$, and $k_{Slw}$ are crop-specific parameters derived from the observational data expressed in Table 1. We conducted curve fitting to $S_{lw}$ calculated from the dry weight of the leaf biomass and the leaf area index based on Ciampitti

et al. (2013a, b) and established a relationship (Figure 3).



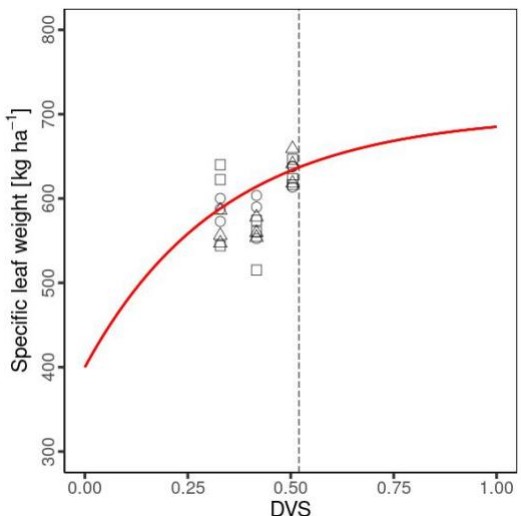

**Figure 3.** Relationships between specific leaf weights and developmental stages. Similar to Fig. 2.

### 2.2.4 Crop height

Crop height ($H_{gt}$) is related to the calculation of evaporation in MATCRO. It assumes that the dependence of the crop height

on $D_{vs}$ is based on Penning de Vries et al. (1989) and is given by

$$H_{gt} = \begin{cases} h_{aa}D_{vs}/D_{vs,flw}, & D_{vs} < D_{vs,flw} \\ h_{aa}, & D_{vs} \geq D_{vs,flw} \end{cases} \tag{34}$$

where $h_{aa}$ is the crop height at flowering, as shown in Table 1.

### 2.2.5 Crop yield

MATCRO calculates the final crop yield, $Y_{ld}$, from the dry weight of the storage organ at maturity ($W_{ear,mt}$) as follows:

$$Y_{ld} = k_{yld}W_{ear,mt}. \tag{35}$$

Here, $k_{yld}$ is the crop-specific parameter (Table 1), which represents the ratio of $Y_{ld}$ to $W_{ear,mt}$. We parameterized $K_{yld}$

based on Ciampitti et al. (2013b).

**Table 1.** Parameters in MATCRO-Maize

| Variable | Value | Units | Description | Source |
|---|---|---|---|---|
| **Crop-specific** | **(maize)** | | | |
| $b$ | 0.04 | mol ($H_2O$) m$^{-2}$ s$^{-1}$ | intercept of the Ball-Berry model | Sellers et al., (1996) |
| $C_{glu,ear}$ | 0.815 | ratio | conversion factor of dry weight from glucose to ear | Penning de Vries et al., (1989) |
| $C_{glu,leaf}$ | 0.871 | ratio | conversion factor of dry weight from glucose to leaf | Penning de Vries et al., (1989) |
| $C_{glu,rot}$ | 0.857 | ratio | conversion factor of dry weight from glucose to root | Penning de Vries et al., (1989) |
| $C_{glu,stm}$ | 0.810 | ratio | conversion factor of dry weight from glucose to stem | Penning de Vries et al., (1989) |
| $D_{vs,rot1}$ | 0.35 | ratio | 1st point of $D_{vs}$ at which the partition pattern to root changes | Penning de Vries et al., (1989) |





| Variable | Value | Units | Description | Source |
|---|---|---|---|---|
| **Crop-specific** | **(maize)** | | | |
| $D_{vs,rot2}$ | 0.72 | ratio | 2nd point of $D_{vs}$ at which the partition pattern to root changes | Penning de Vries et al., (1989) |
| $D_{vs,ear1}$ | 0.37 | ratio | 1st point of $D_{vs}$ at which the partition pattern to ear changes | Parameterized in this study |
| $D_{vs,ear2}$ | 0.6 | ratio | 2nd point of $D_{vs}$ at which the partition pattern to ear changes | Parameterized in this study |
| $D_{vs,flw}$ | 0.52 | ratio | $D_{vs}$ at flowering | Parameterized in this study |
| $D_{vs,lef1}$ | 0.25 | ratio | 1st point of $D_{vs}$ at which the partition pattern to leaf changes | Parameterized in this study |
| $D_{vs,lef2}$ | 0.48 | ratio | 2nd point of $D_{vs}$ at which the partition pattern to leaf changes | Parameterized in this study |
| $f_{stc}$ | 0.35 | ratio | fraction of glucose allocated to starch reserves | Penning de Vries et al., (1989) |
| $h_{aa}$ | 2 | m | crop height at flowering | Penning de Vries et al., (1989) |
| $k_{yld}$ | 0.83 | ratio | ratio of crop yield to dry weight of ear at maturity | Parameterized in this study |
| $k_{Slw}$ | 3 | ratio | parameter that represents the relationship between $S_{lw}$ and $D_{vs}$ | Parameterized in this study |
| $m$ | 4 | ratio | the slope of the Ball-Berry model | Sellers et al., (1996) |
| $G_{dd,m}$ | – | K day | growing degree day at maturity | Parameterized in this study |
| $P_{lef}$ | 0.49 | ratio | partition ratio of glucose to leaf from glucose partitioned to the shoot | Parameterized in this study |
| $P_{rot}$ | 0.25 | ratio | partition ratio of glucose to root | Penning de Vries et al., (1989) |
| $r_{dl,lef}$ | $3.0\times10^{-7}$ | $s^{-1}$ | ratio of dead leaf at harvest | Masutomi et al., (2016) |
| $r_{rt}$ | 0.06 | $m\ s^{-1}$ | growth ratio of root | Penning de Vries et al., (1989) |
| $S_{ln,plt}$ | 0.825 | $g\ m^{-2}$ | specific leaf nitrogen at planting | Parameterized in this study |
| $S_{ln,mx}$ | See Eq. (29) | $g\ m^{-2}$ | maximum specific leaf nitrogen | Parameterized in this study |
| $S_{ln,matu}$ | See Eq. (30) | $g\ m^{-2}$ | specific leaf nitrogen at maturity | Parameterized in this study |
| $S_{lw,mn}$ | 400 | $kg\ ha^{-1}$ | minimum specific leaf weight | Parameterized in this study |
| $S_{lw,mx}$ | 700 | $kg\ ha^{-1}$ | maximum specific leaf weight | Parameterized in this study |
| $T_b$ | 281.75 | K | minimum temperature for development | Osborne et al., (2015) |
| $T_h$ | 315.15 | K | maximum temperature for development | Osborne et al., (2015) |
| $T_o$ | 303.15 | K | optimal temperature for development | Osborne et al., (2015) |
| $z_{rt,mx}$ | 1.5 | m | maximum root depth | Penning de Vries et al., (1989) |
| $\alpha$ | 0.05 | $mol\ mol^{-1}$ | quantum efficiency | Sellers et al., (1996) |
| $\beta_{cj}$ | 0.8 | ratio | GPP transition factor | Lawrence et al., (2020) |
| **Others** | | | | |
| $k_n$ | 0.3 | ratio | vertical distribution of nitrogen | Oleson et al., (2013) |
| $S_1$ | 0.3 | $K^{-1}$ | temperature dependence of $V_{cmax,x}$ | Lawrence et al., (2020) |
| $S_2$ | 313.15 | K | temperature dependence of $V_{cmax,x}$ | Lawrence et al., (2020) |
| $S_3$ | 0.2 | $K^{-1}$ | temperature dependence of $V_{cmax,x}$ | Lawrence et al., (2020) |
| $S_4$ | 288.15 | K | temperature dependence of $V_{cmax,x}$ | Lawrence et al., (2020) |
| $\beta_{ip}$ | 0.95 | ratio | GPP transition factor | Lawrence et al., (2020) |

### 2.3 Model validation

MATCRO can run the simulation both at a point scale and at a global scale. The developed model was validated both at a point scale and at a global scale. For point scale validation, the three model output datasets, LAI, total aboveground biomass,





and yield, were compared with the observation data from the four sites. After confirming the ability of the model to simulate maize growth, two types of validations were conducted at the global scale. First, the simulated yields at the grid cell were compared with the gridded yield data of the Global Dataset of Historical Yields (GDHY) (Iizumi, 2019). Second, the simulated yields at the country and total global levels were compared with the country yield report and global data from the Food and Agriculture Organization (FAOSTAT, 2024). To quantify the model performance, four statistical values were used in this study: the Pearson correlation coefficient (COR), root mean square error (RMSE), relative root mean square error (RRMSE) and normalized mean absolute error (NMAE). RRMSE and NMAE were calculated as follows:

$$RMSE = \sqrt{\frac{1}{n}\sum_{i=1}^{n}(y_i - \hat{y}_i)^2}, \tag{36}$$

$$RRMSE = \frac{RMSE}{\overline{Y}}, \tag{37}$$

$$NMAE = \frac{1}{n}\sum_{i=1}^{n}\frac{|\hat{y}_i - y_i|}{y_i}, \tag{38}$$

where $y_i$ is the actual value, $\hat{y}_i$ is the predicted value, and $\overline{Y}$ is the mean of the actual value.

### 2.3.1 Model validation at a point scale

To validate the model accuracy at a field scale, we used observational data from four sites (Brazil, France, Tanzania, and the USA; Table 2) used in the Agricultural Model Intercomparison and Improvement Project (AgMIP) study (Bassu et al., 2014). We used local daily climate data of precipitation, downwards shortwave radiation, air temperature, wind speed ($P_{rc}$, $R_s$, $T_a$, $U$ respectively), management data ($N_{fert}$ and irrigation regime) and phenological data (planting, flowering, and maturity dates) for model input data at each site. There was no information on the classification of soil texture, so we extracted one grid dataset from the same data used for the global simulation (Section 2.3.2). Annual $CO_2$ data were also taken from the same data used for the global simulation. Climatic data were estimated from the NASA Modern Era Retrospective-Analysis for Research and Applications (MERRA; Rienecker et al., 2011) when measured data were unavailable (Bassu et al., 2014).

**Table 2.** Validation site information in the point-scale simulation

| Country | Site | Latitude | Longitude | Soil type | Sowing date | Hybrid | Total N fertilizer [kg N ha⁻¹] | Irrigation |
|---------|------|----------|-----------|-----------|-------------|--------|----------------------------------|------------|
| Brazil | Rio Verde | 17.52°S | 51.43°W | Geri-Gibbsic Ferralsol | Oct. 22nd 2003 | Pioneer 30K75 | 0 | No |
| France | Lusignan | 46.25°N | 00.07°E | Cambisol | Apr. 26th 1996 | Furio | 255 | Yes |
| Tanzania | Morogoro | 06.50°S | 37.39°E | Haplic Arenosol | Oct. 26th 2009 | TMV1 | 61 | Yes |
| USA | Iowa | 42.01°N | 93.45°W | Gleysols | May 4th 2010 | Golden Harvest GH-9014 | 167 | No |

Notably, air pressure ($P_s$) and specific humidity ($S_h$) were not provided. We used the same data as the global simulation for the soil classification and $P_s$. $S_h$ was converted from $R_h$ using $T_a$ and the vapour pressure. We parameterized $G_{dd,m}$ and



$D_{vs,flw}$ based on $T_a$ and phenological data (sowing, flowering, and maturity dates). $G_{dd,m}$ parameterized for each site is used for the simulations, while the average $D_{vs,flw}$ over the 4 sites is used (0.52 in Table 1). As a result, the mean average errors

were estimated as 4.25 and 7 days for flowering and maturity, respectively (Figure 4). MATCRO was run with these parameters, and then the model output was validated with the observations for the following 3 variables: seasonal change in the LAI, total aboveground biomass, and final yield.

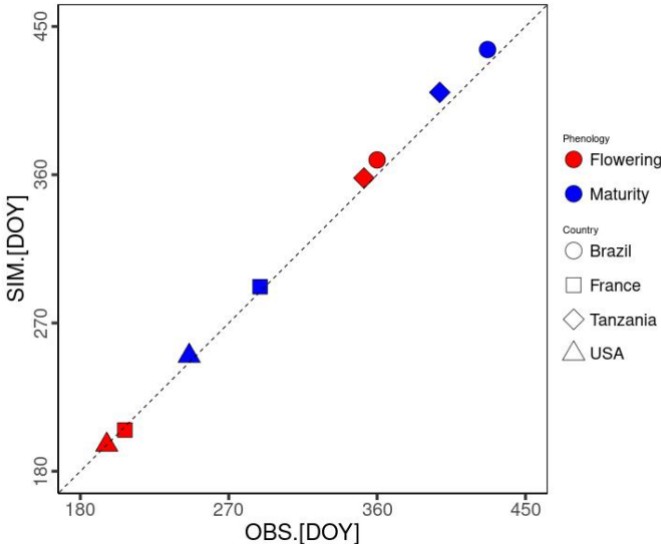

**Figure 4.** Comparison of the flowering and maturity date simulations (SIM on the y-axis) and observations (OBS on the x-axis). DOY represents the number of days from January 1st. Shapes show each site: Brazil (square), France (circle), Tanzania (triangle), and the USA (diamond). The colours indicate the phenological stages: flowering (red) and maturity (blue).

### 2.3.2 Model validation at a global scale

**Simulation settings**

For the global-scale simulation, the model was run at a spatial resolution of 0.5° × 0.5° from 1980–2010 under both rainfed and irrigated conditions. The required input data were as follows. (i) Crop calendar data were from the Global Gridded Crop Model Intercomparison (GGCMI) phase 3 protocol (Jägermeyr et al., 2021). It provides planting and maturity dates for 18 different crops, including maize, separated by rainfed and irrigated systems. We parameterized the average $G_{dd,m}$ at each grid over the period 1980-2010 for the growing season from the planting to maturity dates for each of the rainfed and

irrigated conditions. Both the planting date and the simulated $G_{dd,m}$ were used as the input data for the global-scale simulations. (ii) Water management data (i.e., irrigation regime) from the MIRCA2000 dataset (Portmann et al., 2010). In the case of irrigated conditions, the soil moisture was set to field capacity during the growing season. (iii) $N_{fert}$ from the Inter-Sectoral Impact Model Intercomparison Project (ISIMIP; Volkholz and Ostberg, 2022). It provides the annual nitrogen fertilizer inputs for five canonical crop types, including C4 annual crops for maize. (iv) Soil texture classification from





310 ISIMIP3a protocol soil input data (Volkholz and Müller, 2020). (v) Annual atmospheric $CO_2$ data from the ISIMIP3a

(Büchner and Reyer, 2022). (vi) Six types of daily meteorological for model inputs ($P_s$, $P_{rc}$, $S_h$, $T_{max}$, $T_{min}$, $T_a$, $U$) from the
GSWP3-W5E5 dataset for the ISIMIP3a dataset (Lange et al., 2022). We set the data from (i), (ii), and (iv) as constants
across the simulation period, whereas the data from (iii), (v), and (vi) are variables.

**Analysis**

The simulated final yields in each grid cell under irrigated and rainfed conditions were aggregated by grid cell, country and
global level with the harvested area from MIRCA2000 data (Portmann et al., 2010) via the following equation for each year
from 1981-2010:

$$Yield_{aggregated} = \frac{\sum_{i=1}^{n}(Yield_{i,rf} \times Area_{i,rf}) + \sum_{i=1}^{n}(Yield_{i,irr} \times Area_{i,irr})}{\sum_{i=1}^{n}(Area_{i,rf} + Area_{i,irr})} \tag{39}$$

where $Yield_{aggregated}$ is the aggregated yield with the total grid cells ($n$) in grid cell $i$. $Yield_{rf}$ and $Yield_{irr}$ are the
simulated yields under rainfed and irrigated conditions, respectively, and $Area_{rf}$ and $Area_{irr}$ are the harvested areas from
MIRCA2000 for rainfed and irrigated conditions, respectively.

The model performance was evaluated by comparing its output with the historical yield dataset. The grid cell-level yield
was averaged across a 30-year period and compared with the Global Dataset of Historical Yields (GDHY) (Iizumi, 2019).
The country- and global-level yields were compared with FAOSTAT data (FAOSTAT, 2024) for the average and annual
variabilities over the 30 years. In the comparison at the country level, we focus on the top 20 maize-producing countries that
account for more than 85% of total maize production.

We focused on two perspectives for validation: (i) the ability of the model to capture the spatial distribution of yield in
both low- and high-producing countries and (ii) the ability of the model to reproduce the climatic effect reflected in the
interannual variability at the country and global scales. The first perspective was analysed using NMAE to quantify model
error for both the global yield and the yield of the top 20 producing countries. The 30-year average yields were also
compared on the basis of the statistics of COR, RMSE, and RRMSE to confirm the accuracy. The second perspective was
analysed via the COR of the detrended deviation between the simulated and FAOSTAT yields to assess the interannual
variability.


**The effects of photosynthesis and N fertilizer**

In addition to the yield comparison, we analysed the effect of nitrogen fertilizer ($N_{fert}$) on maize yield, as it is a key
determinant of crop yield. This analysis compared both FAOSTAT data and simulated data from $N_{fert}$ for a 30-year average
with simple linear regression. We also conducted two tests to quantify the effects of the $N_{fert}$-related function and
parameters as follows: (i) Eq. (27) during the vegetative stage is derived from Drouet and Bonhomme (2004), defined as
"test $S_{ln}$-$V_{cmax}$", was changed to:





$$V_{cmax}(0) = 36.8 * \left\{\frac{2}{1+\exp[-2.45*(S_{ln}-0.27)]} - 1\right\}, D_{vs} < D_{vs,flw} \tag{40}$$

and (ii) $S_{ln,plt}$ from 0.825 (Table 1) to 0.5 (defined as "test $S_{ln,plt}$").

## 3 Results

### 3.1 Point-scale simulations

A comparison of the time series changes in the LAI at each experimental site is shown in Figure 5. In general, MATCRO-Maize captured the increasing trend towards flowering time and then decreasing trend towards the end of maturity. Especially during the vegetative stage ($D_{vs} < D_{vs,flw}: 0.52$), the simulated LAI showed relatively good agreement. However, the simulated LAI was notably underestimated in Brazil and France immediately before the reproductive stage (near the dashed black line in Fig. 5).

Figure 6 compares the time series of total aboveground biomass between the simulated and experimental data. Except for Tanzania, MATCRO-Maize accurately estimated the increasing trend of total aboveground biomass towards maturity, although the simulated biomass in Brazil was underestimated at maturity. The simulated total aboveground biomass in Tanzania increased until maturity, while the observations gradually decreased towards maturity time (Fig. 6 (c)).

Figure 7 compares the 1:1 line between the simulated and experimental data for the seasonal LAI (Fig. 7 (a)), seasonal total aboveground biomass (Fig. 7 (b)), and harvested yield (Fig. 7 (c)). The LAI underestimation in France and Brazil (Fig. 5) could also be seen with a large RMSE, which is approximately 50% of the average LAI across all observational values at 3 sites except for Tanzania, although overall, the comparison was statistically significant (p value < 0.01), with a COR of 0.762. The comparison of total aboveground biomass was statistically significant (p value < 0.001), with a COR of 0.895, although the RMSE was 3,628.3 [kg ha⁻¹], which corresponds to approximately 35% of the average of all observed total aboveground biomass. While the comparison of the final crop yield was statistically significant (p value < 0.01), there was a relatively low COR compared with the LAI and total aboveground biomass due to the small sample size (N=4) and the overestimation for Tanzania. The RMSE was 2,575.0 [kg ha⁻¹], which is approximately 30% of the average observational yield at all the sites.





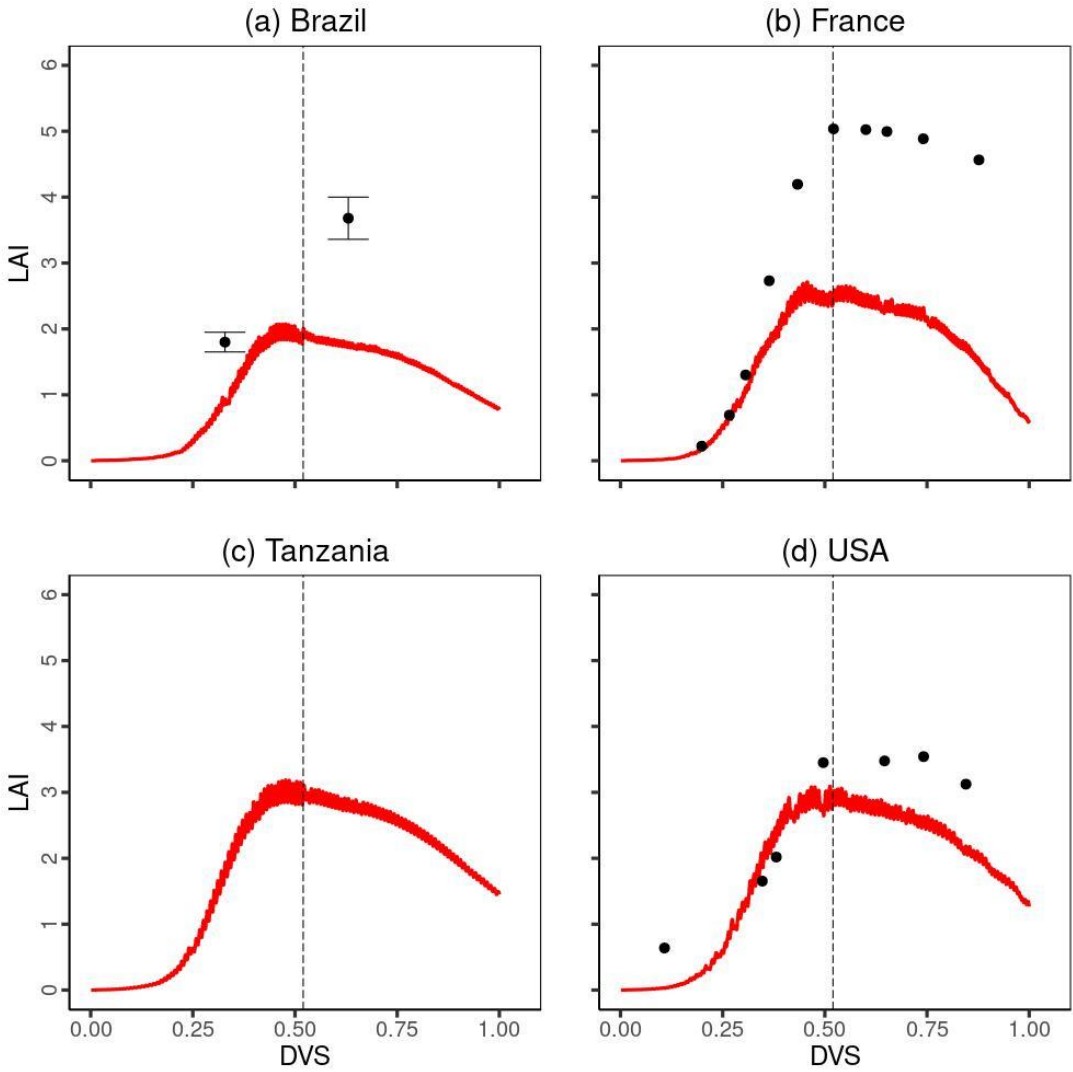

**Figure 5.** Temporal evaluation of leaf area index (LAI) simulated by MATCRO-Maize (red line) at each site: (a) Brazil, (b) France, (c) Tanzania and (d) the USA across the developmental stage ($D_{vs}$). The observation data in each site is shown by black point. Notably, there were no observational data in Tanzania. The error bars were provided only for Brazil. The dashed black line shows the flowering time.






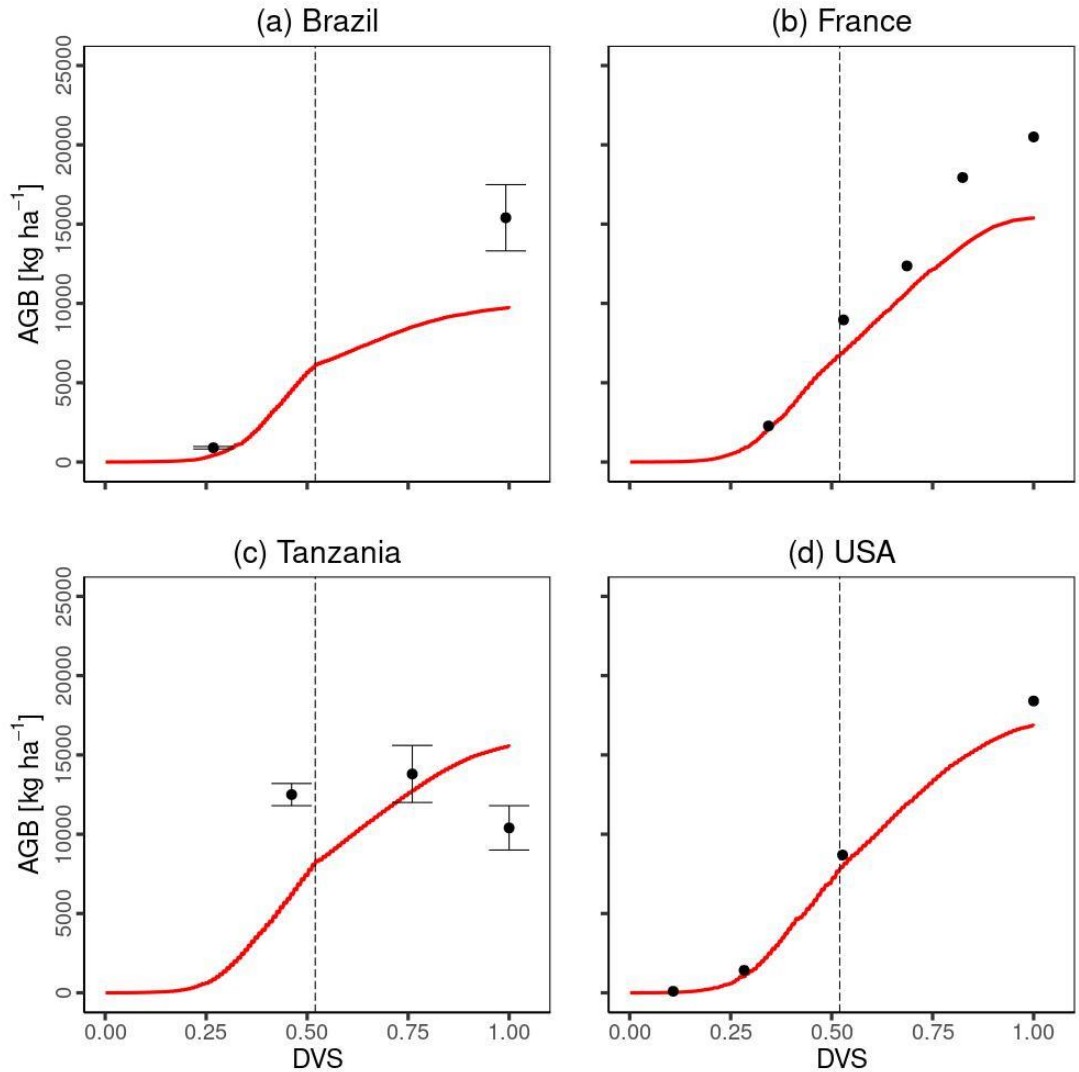

**Figure 6.** Temporal evaluation of total aboveground biomass (AGB) simulated by MATCRO-Maize (red line) at each site: (a) Brazil, (b) France, (c) Tanzania and (d) the USA across the developmental stage ($D_{vs}$). The observation data in each site is shown by black point. The error bars were only provided for Brazil and Tanzania. The dashed black line shows the flowering time.




**Figure 7.** Statistical comparison (COR, RMSE, and RRMSE) of (a) LAI, (b) Total aboveground biomass, and (c) Yield. The x-axis (OBS) represents the observational data, and the y-axis (SIM.) is the simulated data. Shapes show each site: Brazil (square), France (circle), Tanzania (triangle), and the USA (diamond). Notably, there was no observed LAI in Tanzania. The symbols ***, **, indicate p values <
0.001 and 0.01, respectively.

**3.2 Global-scale simulations**

A comparison of the global distributions is shown in Figure 8 (simulation: Fig. 8(a); GDHY dataset: Fig. 8(b)). While the overestimation could be seen mainly in tropical regions, the simulated yield could capture high-yielding regions, including the Corn Belt in the United States and the northern part of China. Temporal changes in the global yield across 30 years
indicated that although the global yield had an NMAE of 0.67, indicating a simulation error of 67% with respect to the



average FAO yield, the comparison of the interannual variability between the simulations and observations was statistically significant (p value < 0.01), with a COR of 0.549 (Figure 9). For the top 20 producing countries, MATCRO-Maize also tended to overestimate the yield in terms of the annual yield (Figure 10) and the average yield over a 30-year period (Figure 11). The overestimation was strong in Egypt, where the simulated yield was approximately four times greater across 30 years.

In terms of interannual variability, half of the 20 countries were statistically significant, with p values < 0.001 for 6 countries, < 0.01 for 2 countries, and < 0.05 for 2 countries (Fig. 10). The 30-year average comparison was also statistically significant (p value < 0.01), with a COR of 0.58, although the RMSE was 4,007.7 [kg ha⁻¹], which is almost the same as the average yield of the top 20 maize-producing countries (Fig. 11).

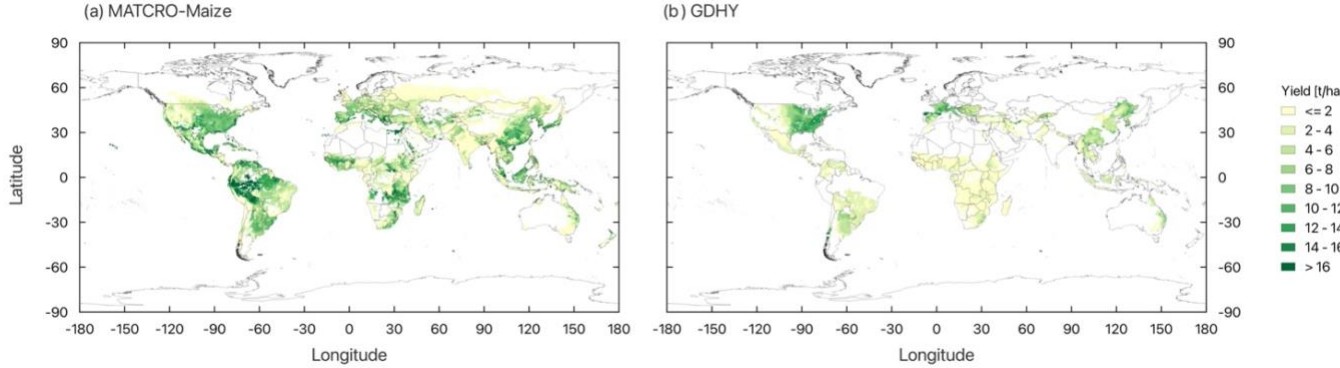

**Figure 8.** Global distribution of the 30-year average (1981-2010) maize yield by (a) simulations from the MATCRO-Maize and (b) the GDHY dataset. The yield is aggregated based on the harvested area from MIRCA2000.

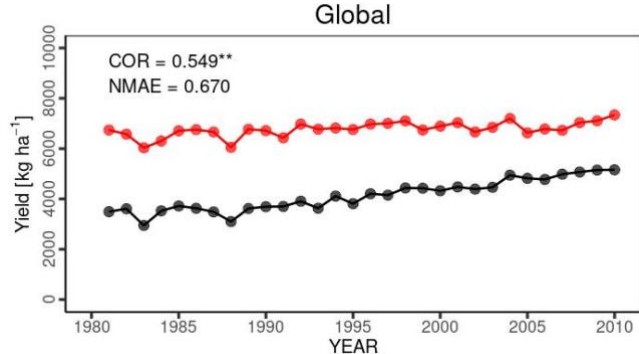

**Figure 9.** Interannual variability in global maize yield from 1981 to 2010 for our simulation (red circles) and FAOSTAT (black) yields. COR represents the correlation coefficient of interannual variability. NMAE means normalized mean absolute error. Asterisks ** indicate p value < 0.01.



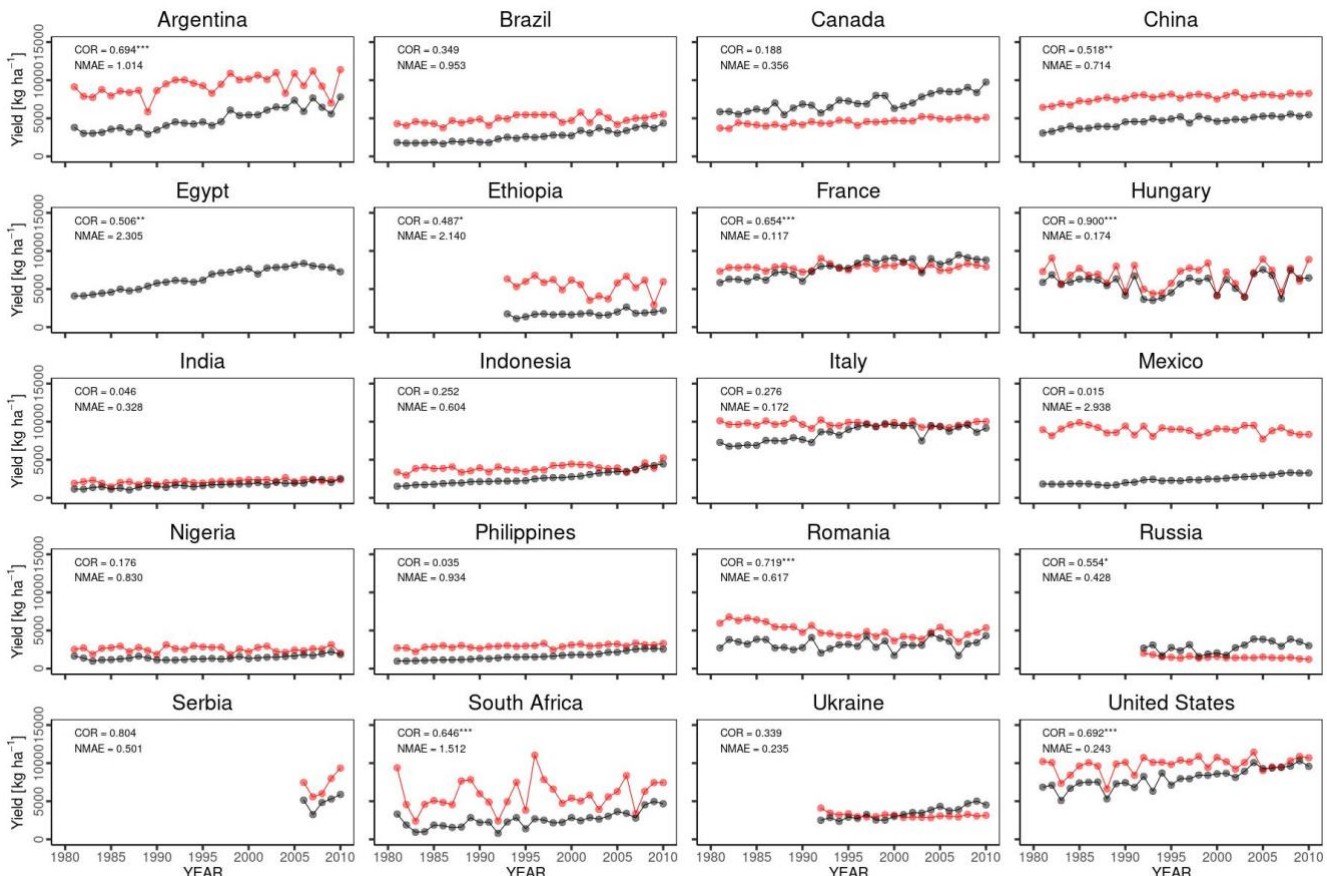

**Figure 10.** Comparison of interannual variability for the top 20 maize-producing countries. Similar to Fig. 9. Notably, the simulated yield in Egypt is not shown as it extends beyond the range of the y-axis. The symbols ***, **, and * indicate p values < 0.001, 0.01, and 0.05, respectively.

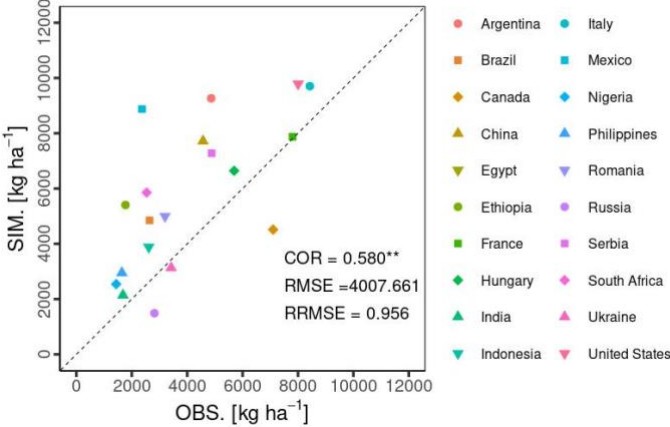

**Figure 11.** Accuracy of the 30-year average of the simulated yield (SIM) to the observed yield (OBS from FAOSTAT data) for the top 20 countries. Notably, the Egypt data points are not shown as exceeding the range of the y-axis. Asterisks ** indicate a p value < 0.01.




Figures 12 (a) and (b) show the comparisons based on $N_{fert}$ for each FAOSTAT and simulated yield, respectively. MATCRO has a strong $N_{fert}$ effect on the yield reflected in the steep slope of the regression line with relatively similar intercepts (slope: 27.6 and 47.3; intercept: 1,701.0 and 2,187.4 for FAOSTAT and MATCRO-Maize, respectively). This effect was scarcely alleviated by the intentionally changed setting (Fig. 12 (c), (d)), mainly because of the effect of Egypt as an outlier. Figure 13 shows the same comparison as Fig. 12 without Egypt. The slope was similar to that of FAOSTAT (33.8 for FAOSTAT and 30.35 for MATCRO-Maize), whereas the intercept was approximately 2.5 times greater (1,310.6 for FAOSTAT and 3,247.0 for MATCRO-Maize) (Fig. 13 (a), (b)). The changed equation and parameter partly alleviated the large intercept, although the slope became slightly smaller than that of FAOSTAT (Fig. 13 (c), (d)).

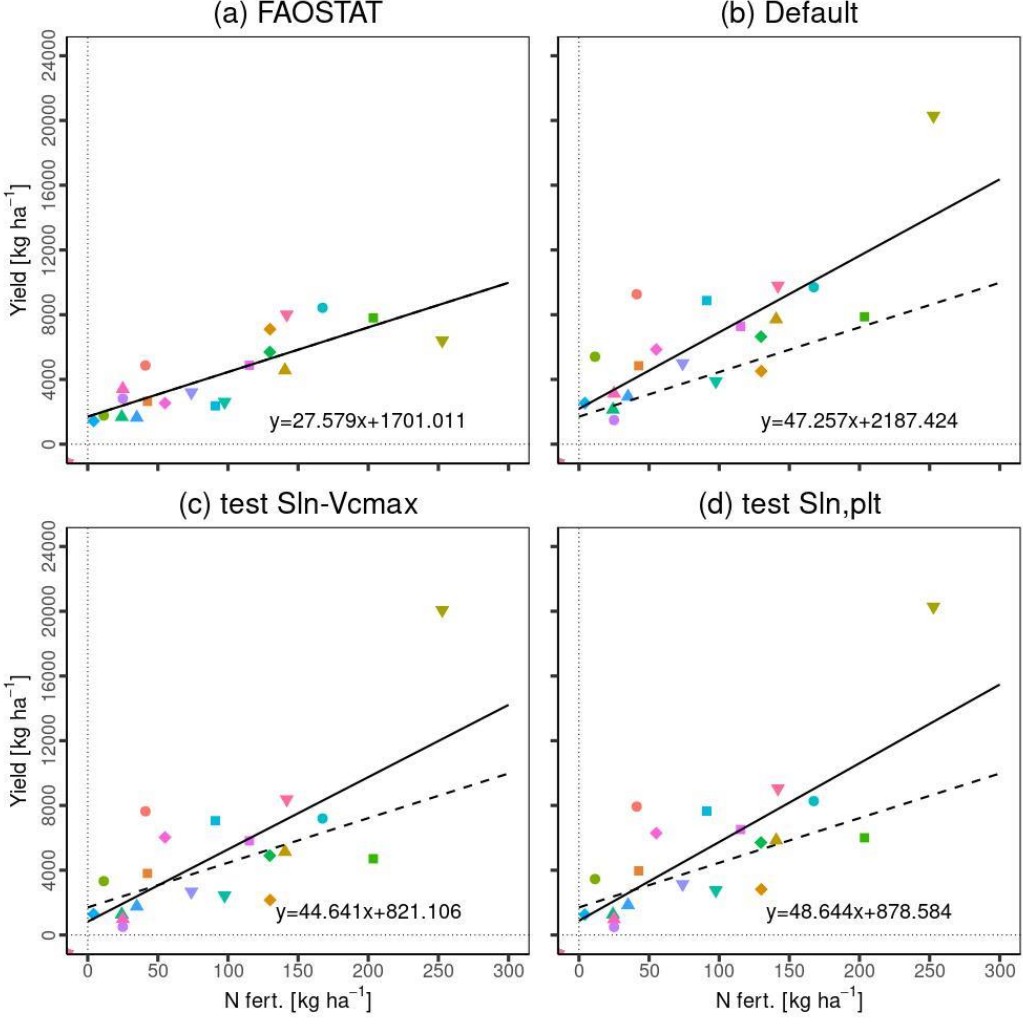

**Figure 12**. $N_{fert}$ impact on yield of (a) FAOSTAT, (b) simulated yield with the original setting (Default), (c) simulated yield with the changed $S_{ln}$-$V_{cmax}$ relationship (test Sln-Vcmax), (d) simulated yield with the changed parameter related to the $D_{vs}$-$S_{ln}$ function (test Sln, plt). $N_{fert}$ (N fertilizer) and country yield were averaged across 30 years for each country. The legends are the same as those in Fig. 11. The dashed lines in (b), (c), and (d) indicate the regression lines in (a).





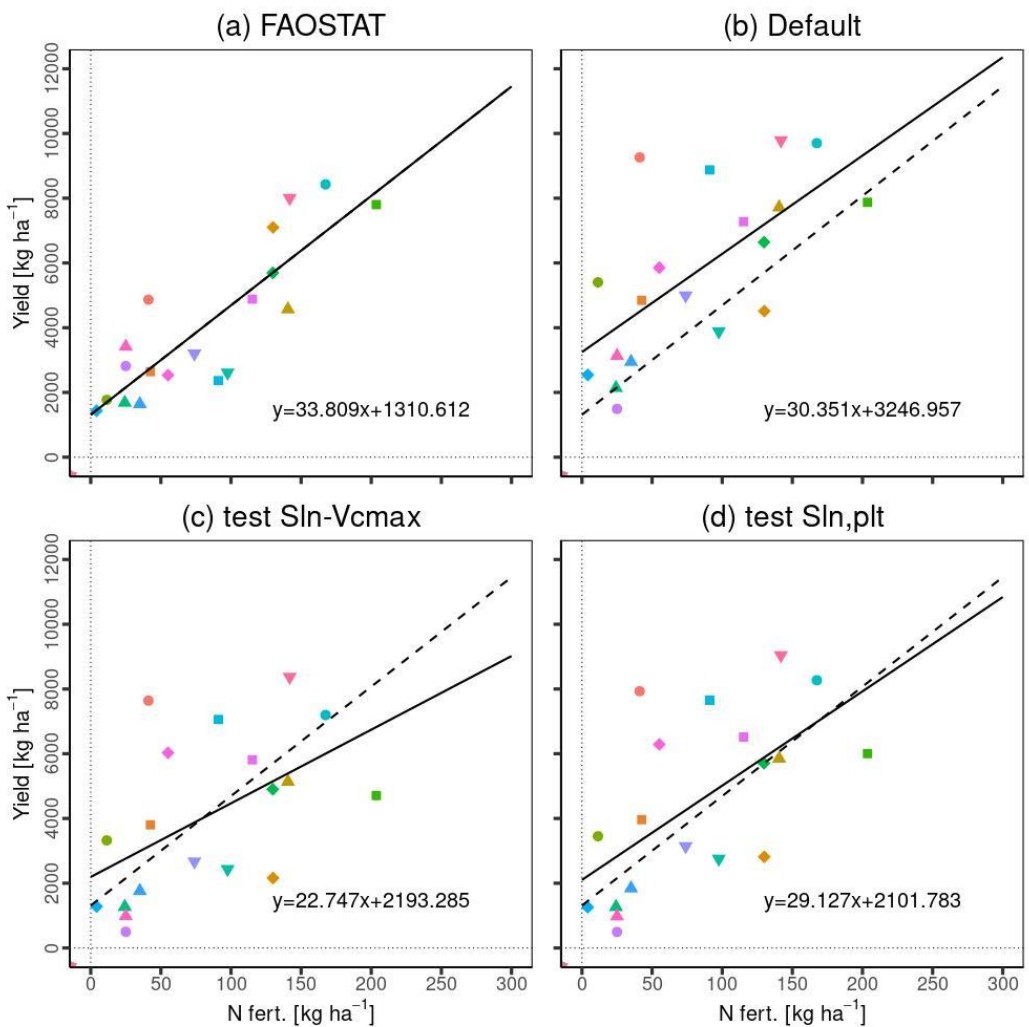

**Figure 13.** $N_{fert}$ impact on yield. Similar to Fig. 12, but without Egypt.

## 4 Discussion

### 4.1 Point-scale simulations

The simulated LAI, total aboveground biomass, and harvested yield were statistically significant at the point scale (Fig. 7), indicating that the MATCRO-Maize model could simulate maize growth and yield relatively well. However, there were some discrepancies between the simulations and observations, such as the underestimation of the LAI in Brazil and France, the underestimation of the total aboveground biomass in Brazil, and the different growth trends of the total aboveground biomass in Tanzania.



The reason for the underestimation of the LAI in France might be related to the effect of plant density, which is not currently considered in MATCRO. The actual plant density [plants m$^{-2}$] at each site was 9.5 (France), 7.5 (USA), 6.6 (Brazil), and 9.5 (Tanzania) (Bassu et al., 2014). Some studies have shown that LAI trends are affected primarily by the plant density factor relative to $N_{fert}$ and hybrids (Boomsma et al., 2009; Ciampitti et al., 2013a; Ciampitti and Vyn, 2011). This may be the reason for the underestimation that MATCRO could not reproduce the trends driven by plant density, although other important factors (e.g., management practices, climatic conditions), which are quite different from each site in the literature, would also affect crop growth variables, including the LAI.

Both the underestimation of the LAI and total aboveground biomass in Brazil were probably caused by the field experimental conditions of $N_{fert} = 0$, given its effect on crop growth in MATCRO. The reason for the lack of fertilization in the field experiment was that sufficient N was released by organic matter mineralization (Bassu et al., 2014), which was not considered in the model. Moreover, $N_{fert}$ directly affects $S_{ln}$ in MATCRO, with an increasing trend towards flowering and then a decreasing trend towards maturity (Fig. 1). $S_{ln}$ is related to $V_{cmax25}(0)$, which in turn affects the photosynthesis calculation (Section 2.1 and Section 2.2.2). In particular, during the reproductive stage, we used Eq. (27), which results in a low $V_{cmax25}(0)$ under low $S_{ln}$ due to the more gradual slope of the curve compared with the vegetative stage (1.41 for the reproductive stage, and 2.9 for the vegetative stage, in Eq. (27)), indirectly leading to low biomass accumulation through photosynthesis. This could be attributed to the underestimation of total aboveground biomass at maturity (Fig. 6 (a)). For underestimation of the LAI, low leaf biomass accumulation, which is derived from the same mechanism, would be the reason considering the calculation process of the LAI in MATCRO. The LAI is determined by the division of the leaf biomass weight by $S_{lw}$, which depends on $D_{vs}$. Because $S_{lw}$ is calculated from the same parameter at all sites (Eq. (33) and Fig. 3), leaf weight is the factor that causes differences between sites, leading to the underestimation of the LAI in Brazil. Therefore, the condition of $N_{fert} = 0$ might be the reason for both underestimations.

One possible reason for the difference in the growth trend of biomass in Tanzania might be related to growing season length. The cultivar used in Tanzania was a short season type with 99 days of observed growing season length, whereas the cultivars at other sites were medium or long season type with lengths ranging from 122 to 173 days (Bassu et al., 2014). Capristo et al. (2007) reported that, compared with medium- and long-season cultivars, short-season cultivars presented the lowest biomass accumulation from flowering to maturity, which was reflected in the observed biomass (Fig. 6 (c)). This might indicate that the trend of biomass accumulation differs across growing season types, although other factors, such as climatic conditions or biotic stresses, could also affect accumulation. While MATCRO considers the growing season length as $G_{dd,m}$ to judge the harvesting time, this does not mean that MATCRO could capture the difference in trends due to growing season types, possibly leading to the gap between the simulations and observations shown in Tanzania.





## 4.2 Global-scale simulations

A comparison of the global distribution of maize yield revealed that MATCRO-Maize could capture the distribution of high-

yield regions but could not capture the yield in tropical regions (Figure 8). Similar overestimations in tropical regions have also been reported in other global models, possibly because of the lack of representation of extreme weather or crop pests (Lombardozzi et al., 2020; Osborne et al., 2015).

Notably, MATCRO-Maize tended to overestimate the absolute values for both the total global yield and the yields of the top 20 countries, as reflected in the NMAE and RMSE values (Figures 9, 10, and 11). The simulated total global yield is

determined mainly by the yield of the top 3 maize-producing countries, the United States, China, and Brazil, which have large cultivated areas (Table 3). All three countries' yields were overestimated, where the simulated yields were approximately 1.2, 1.7, and 1.8 times greater for the 30-year averages in the United States, China, and Brazil, respectively, leading to overestimation of the total global yield. Such overestimations in the main producing countries, especially in China and Brazil, are also observed in other global crop models (Von Bloh et al., 2018; Osborne et al., 2015; Schaphoff et al.,

2018). This might indicate that there are factors that are important for determining yields but are not considered in most crop models.

For the top 20 producing countries, the overestimation was strong in Egypt, with an approximately fourfold greater simulated yield than that of FAOSTAT. This overestimation might be caused by the irrigated conditions in all grids in Egypt. Under manually changed rainfed conditions, crop growth in Egypt in the model was almost not simulated because of the

inhibited photosynthesis rate caused by strong water stress. Under irrigated conditions, this strong water stress was alleviated. In addition, the radiation in Egypt was consistently strong throughout the growing period, and $N_{fert}$ was highest among the top 20 countries across the 30 years simulated, increasing from approximately 180 kg ha$^{-1}$ in 1980 to 360 kg ha$^{-1}$ in 2010. This caused the colimited photosynthesis rate to be high (Eq. (4)) across the growing seasons, leading to marked overestimation.

Although the simulated yield has the large error in terms of the absolute value, the comparison of the 30-year average yield was statistically significant, with a COR of 0.58 (p value < 0.01) and an RMSE of 4,008 kg ha$^{-1}$ (Fig. 11), showing the ability to capture the spatial distribution of the yield both in low- and high-producing countries from the first perspective of the comparison (Section 2.3.2). This result was comparable with the similar result of another model: LPJ-GUESS (Olin et al., 2015), with a COR of 0.46 and an RMSE of 4,300 kg ha$^{-1}$ (Table 4), although the targeted countries were different (top 20

producing countries for MATCRO-Maize, whole countries for LPJ-GUESS).

In terms of interannual variability from the second perspective, the total global yield and approximately one-third of the top 20 producing countries were statistically significant, with p values < 0.01 (Figs. 9 and 10), indicating that MATCRO-Maize could reproduce the climatic effect globally to some extent. This might also be supported by the similar comparisons of other global crop models in terms of statistics (Table 4), although it is difficult to simply compare the statistical values

between the models owing to the differences in periods, input data, and methods for detrending and aggregating the yield.



The COR of interannual variability for total global yield in MATCRO-Maize was in the range of those of the other models (0.55; 0.42~0.89, respectively). For the top 20 countries, almost all the COR values also ranged between those of the other models. Therefore, these comparisons from two perspectives might indicate that MATCRO-Maize could yield reasonable results.


**Table 3.** Maize cultivated land area for 20 major producer countries from MIRCA2000 (Portmann et al., 2010).

| Country | Total area [ha] | Rainfed area [ha] | Irrigated area [ha] |
|---|---|---|---|
| Argentina | 3,248,715.9 | 3,147,580.7 | 101,135.3 |
| Brazil | 11,223,262.5 | 11,120,154.9 | 103,107.6 |
| Canada | 1,364,585.3 | 1,328,206.2 | 36,379.1 |
| China | 24,376,805.2 | 11,615,190.0 | 12,761,615.2 |
| Egypt | 827,766.1 | 0.0 | 827,766.1 |
| Ethiopia | 1,172,231.1 | 1,084,795.6 | 87,435.5 |
| France | 3,128,401.0 | 2,257,380.0 | 871,021.0 |
| Hungary | 1,057,610.7 | 1,052,622.6 | 4,988.1 |
| India | 6,294,770.9 | 4,833,685.9 | 1,461,085.0 |
| Indonesia | 3,479,825.7 | 3,135,443.9 | 344,381.8 |
| Italy | 1,322,692.9 | 534,281.4 | 788,411.5 |
| Mexico | 7,459,039.5 | 5,852,617.4 | 1,606,422.1 |
| Nigeria | 3,686,757.3 | 3,667,564.5 | 19,192.8 |
| Philippines | 2,590,081.0 | 2,590,081.0 | 0.0 |
| Romania | 3,139,981.1 | 3,016,990.5 | 122,990.6 |
| Russia | 4,206,747.0 | 3,594,403.2 | 612,343.9 |
| Serbia | 1,074,614.2 | 1,062,985.8 | 11,628.4 |
| South Africa | 3,060,053.5 | 2,930,208.2 | 129,845.4 |
| Ukraine | 3,382,783.5 | 3,194,146.2 | 188,637.3 |
| United States | 31,307,667.3 | 26,508,600.7 | 4,799,066.7 |

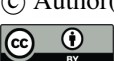

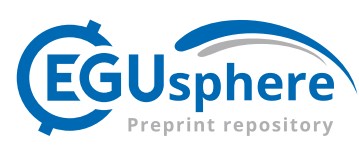


**Table 4.** Statics of model simulation accuracy of the MATCRO-Maize and other crop models. Notably, the asterisks for GGCMI phase I indicate the p values: *** for p values < 0.001, ** for p values < 0.05, * for p values < 0.1, whereas those of LPJmL4 and MATCRO-Maize indicate the p values: *** for p values < 0.001, ** for p values < 0.01, * for p values < 0.05.

| | References | Period | COR of interannual variability | | | | | | |
| --- | --- | --- | --- | --- | --- | --- | --- | --- | --- |
| | | | Global | USA | China | Brazil | Mexico | France | Argentina |
| MATCRO-Maize | - | 1981–2010 | 0.549** | 0.692*** | 0.518** | 0.349 | 0.015 | 0.654*** | 0.694*** |
| JULES-crop[1] | Osborne et al., 2015 | 1961–2008 | 0.48 | 0.43 | 0.12 | 0.12 | 0.061 | 0.52 | 0.57 |
| LPJmL4[2] | Schaphoff et al., 2018 | 1981–2010 | – | 0.675*** | 0.676*** | 0.169 | -0.124 | -0.331 | 0.717*** |
| LPJmL5[3] | Bloh et al., 2018 | 1981–2010 | – | 0.686*** | 0.641*** | 0.0591 | 0.0618 | 0.461* | 0.650*** |
| GGCMI phase 3[4] | Jägermeyr et al., 2021 | 1981–2015 | – | 0.817 | 0.245 | 0.029 | – | 0.649 | 0.727 |
| GGCMI phase 1[5] | Müller et al., 2017 | 1982–2006 | 0.42**~0.89*** | 0.89 | 0.75 | 0.66 | 0.85 | 0.87 | 0.85 |

| | References | Period | COR of interannual variability | | | | | | |
| --- | --- | --- | --- | --- | --- | --- | --- | --- | --- |
| | | | Romania | South Africa | India | Italy | Hungary | Indonesia | Ukraine |
| MATCRO-Maize | - | 1981–2010 | 0.719*** | 0.646*** | 0.046 | 0.276 | 0.900*** | 0.252 | 0.339 |
| JULES-crop[1] | Osborne et al., 2015 | 1961–2008 | 0.32 | 0.41 | 0.34 | 0.34 | 0.33 | 0.065 | – |
| LPJmL4[2] | Schaphoff et al., 2018 | 1981–2010 | – | 0.711*** | -0.22 | – | – | 0.124 | -0.046 |
| LPJmL5[3] | Bloh et al., 2018 | 1981–2010 | – | 0.667*** | 0.496** | – | – | -0.163 | 0.152 |
| GGCMI phase 3[4] | Jägermeyr et al., 2021 | 1981–2015 | – | – | – | – | – | – | – |
| GGCMI phase 1[5] | Müller et al., 2017 | 1982–2006 | 0.90 | 0.91 | 0.76 | 0.76 | 0.90 | 0.42 | 0.61 |

| | References | Period | 30-year averaged yield | |
| --- | --- | --- | --- | --- |
| | | | COR | RMSE [kg ha⁻¹] |
| MATCRO-Maize | - | 1981–2010 | 0.580** | 4,008 |
| LPJ-GUESS[6] | Olin et al., 2015 | 1996–2005 | 0.46 | 4,300 |

[1] Countries-level comparison was conducted for 12 countries, which were detrended only for observation. p values are not shown.

[2,3] Countries-level comparison was conducted for the top 10 producing countries, which were detrended via a 5-year moving average.

[4] Twelve global gridded crop models were used. The COR shown here is the ensembled mean value for the 5 largest producing countries after detrending. p values are not shown.

[5] Fourteen global gridded crop models were used. The COR of the global yield shown here is the minimum and maximum value, except for one nonsignificant correlation with the default setting. The COR of each country is the best correlation among the 14 models, including 3 different settings with statistical significance (p values are not shown). For both the global and country-level comparisons, a 5-year moving average was used to remove trends.

[6] The 10-year average comparisons included all countries. p values are not shown.




### 4.3 Model limitations

The limitation of the current MATCRO-Maize is the strong $N_{fert}$ effect shown in the validation (the site in Brazil for the point scale) and comparison based on the $N_{fert}$ (Figures 12 and 13). In the model, $N_{fert}$ has the direct relationship with $S_{ln}$ (Eq. (28)) and consequently affects $V_{cmax25}(0)$ through the function $S_{ln}$-$V_{cmax25}(0)$ (Eq. (27)). Therefore, the strong $N_{fert}$ effect is caused by either the former, the latter, or both processes. Few studies have explicitly shown time series changes in $S_{ln}$ and $S_{ln}$-$V_{cmax}$ relationships from experiments. We used some of them to establish the functions shown in Eqs. (27) and

(28) (Section 2.2.2) at this stage, resulting in a strong $N_{fert}$ effect in the model. However, the intentional experiment indicated that the changed relationships could partly reproduce the adequate effect, which was observed in the FAOSTAT yield. This might mean that the established functions include a degree of uncertainty, and if we establish robust relationships based on other experimental data under more comprehensive conditions, it might be possible to improve the model in terms of the $N_{fert}$ effect, leading to a more accurate simulation of maize yield.

In this study, we applied identical parameters to simulate the global yield across all grid cells and throughout the years without considering cultivar differences. As mentioned in Section 3.1.2, the trend of biomass accumulation would differ across growing season types. Moreover, in major producing countries, such as the United States and China, some studies have shown that there is genetic gain in terms of maize yield (Cooper et al., 2014; Duvick et al., 2003; Liu et al., 2021). Such cultivar differences and long-term genetic improvements are not included in the current MATCRO-Maize. This finding

indicates that the generic parameterization used in the model would be simple in accounting for the diversity of crop cultivars (Lombardozzi et al., 2020), partly leading to a gap between the simulations and observations, which is recognized as a limitation of the global model (Osborne et al., 2015).

    In addition, other important factors that are not considered in the current MATCRO also affect crop growth and final yield. These factors include biotic stresses (e.g., diseases, pests) and detailed management practices (e.g., plant density, as

mentioned in Section 4.1). Further improvement to incorporate such factors with reliable $N_{fert}$-related functions could be needed to contribute to more accurate simulations and contribute to studies on the interaction between climate and agriculture.

### 5 Conclusions

    We developed a process-based crop model for maize yield estimation, called MATCRO-Maize, by incorporating C4 leaf-

level photosynthesis and some crop-specific parameters into MATCRO. The model was first validated at the point scale, showing a somewhat reasonable accuracy considered with insufficient field-based information for parameterization. The calibrated parameters were set from point-scale experimental data and used uniformly in the global-scale simulation. MATCRO-Maize could represent the spatial distribution well and showed reasonable responses to climatic variability, where the results were comparable with those of other studies in terms of statistics. The strong nitrogen fertilizer effect was one of



the MATCRO limitations, while the established functions related to nitrogen fertilizer in the model have a degree of uncertainty. Further experimental data under more comprehensive conditions might improve the model. Overall, MATCRO-Maize could contribute to climate effect studies through its ability to be integrated with the LSM for crop growth and the interactions between climate and agriculture.

*Code and data availability.*

The source code used in this study is archived at https://doi.org/10.5281/zenodo.14869445 (Nagata et al., 2025).

*Author contributions.*

TK supervised this study and acquired the funding. YM developed the model source code and supervised this study. MN adjusted the source code for this study, performed the model simulations, and analysed the model output. MN prepared the original draft, which was supervised by TK, YM, and AY. The final manuscript was written by MN and approved by all the
authors.

*Acknowledgements.*

This research was funded by JSPS KAKENHI grant number JP 22k18497. This study was partly supported by the Environment Research and Technology Development Fund (JPMEERF20202005) of the Environmental Restoration and Conservation Agency (ERCA) and JSPS KAKENHI Grant Number JP23H00351.
We thank all the contributors. Dr. Jean-Louis Durand at the French National Research Institute for Agriculture, Food, and Environment (INRAE) provided the observational data at the four sites, which we used to validate our model at the point scale.

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
