# Peer review of "Development of the global maize yield model MATCRO-Maize version 1.0"

_EGUsphere, 2025_

## Author Comment (AC1)

*Dear RC1,*

*Thank you very much for your thorough efforts to highlight these important aspects, which has helped us clarify the model description.*

***First, we will reply for your major concern:***
"1. The Nfert problems suggest to me that MATCRO is very deficient in having, or totally lacking, a soil organic matter module and lacking in plant N balance. The authors need to come clean on this and state in the text that MATCRO either has or lacks a plant N balance module and soil N module. Soils were adequately described for water balance processes, but there was no mention of soil organic matter or soil N mineralization. Basing SLN and Vcmax on Nfert is a very limiting approach and suggests that soil N supply is ignored. The authors need to state those limitations and improvements even more strongly."

*Reply: We thank the reviewer for emphasizing the importance of nitrogen balance module. We agree with these points and acknowledge that our earlier description did not sufficiently state this limitation. This model using the simplification of nitrogen representation in the leaf, where fertilization rate implicitly affects photosynthetic parameters (specific leaf nitrogen and maximum Rubisco capacity) without simulating soil nitrogen mineralization or plant nitrogen balance. We will clarify this in Section 4.3 of the manuscript:*

*"The model currently lacks explicit simulation of plant nitrogen balance. Nitrogen supply is represented by the effect of fertilizer rate to specific leaf nitrogen (SLN), which represents the leaf nitrogen content across a wide range of fertilizer levels (Muchow, 1988). This relationship is then used to adjust photosynthetic capacity (Vcmax). While this simplification enables global-scale application, it limits the model ability to capture nitrogen balance effects on yield. Future development could involve coupling MATCRO with a mechanistic soil nitrogen module to simulate soil nitrogen mineralization and dynamic plant nitrogen balance, thereby improving the model's capacity to represent nitrogen effects under diverse soil and management conditions."*

"2. The authors use the term validation. Are Figures 5-10 independent data? But if so, then what data did you use for calibration? Was calibration only from literature for individual parameters? Need to indicate what data was used for calibration as that was never stated. Otherwise, readers will suspect you used this data for calibration."

*Reply: Thank you for raising this important point. Pardon us for the confusion as our manuscript did not clearly state calibration and evaluation. Figures 5-10 are independent data. Calibration was limited to partitioning parameters (e.g. leaves, ear; and specific leaf weight toward developmental stage) derived from studies of Ciampitti et al. (2013a,b), since seasonal partitioning data were not available in the experimental datasets (Bassu et al., 2014). The comparisons of LAI, biomass, and yield in Figures 5– 7 for point scale, as well as yield in Figures 8–10 at the global-scale (spatial distribution from GDHY and country global-level from FAO), were used only for evaluation and not for calibration. However, we will replace the term "validation" with*

*"evaluation" to avoid misunderstanding and explicitly state which data were used for calibration and which were held out for evaluation.*

"3. The statistics of Figures 5, 6, and 7 indicate quite poor performance of MATCRO-Maize and correlation statistics are weak tests. Can we recommend a model that performs that poorly for use by global gridded teams? Figure 7A and 7B are based on LAI and crop mass over time, but the high correlations there are misleading because of auto-correlation effects of time-series data (give high correlation because it uses time-series values)."

*Reply: We agree that correlation based on time-series data lead to autocorrelation (e.g. LAI and biomass over time). To avoid misleading results, we will remove Figure 7A and 7B from the performance test (for LAI and total aboveground biomass). While site-level optimization would demand detailed information on local varieties and management, our approach emphasizes capturing global yield patterns. Hence, we can recommend this model for global scale simulation with the current parameterization. The model shows a moderate correlation at the global scale correlation (R = 0.54) and top 20 major producing countries (R = 0.58) compared with FAO data despite of data limitations. We are confidence that MATCRO-Maize is informative for global yield pattern, while acknowledging its limitations.*

**About your concern on technical correction:**

1. "l. 83 – Says C-3 here. Should be C-4."

*Reply: You are correct, thank you for noticing this error. We have corrected the text accordingly: "MATCRO-Maize calculates net carbon assimilation for the entire canopy (An) via the big-leaf model (Dai et al., 2004), where C4 leaf-level photosynthesis is separately calculated for sunlit and shaded leaves from the coupled photosynthesis–stomatal conductance model (Collatz et al., 1992)."*

2. "L 101 – why bother with "co-limited" photosynthesis? That is a C-3 hold-over and probably does not apply to C-4."

*Reply: Pardon us for the confusion, Thank you for pointing this out. We understand that "co-limited" photosynthesis mainly refers to C3 and is not appropriate for C4, hence we have removed it from the manuscript.*

3. "l. 120 – Is "$0.7\mu$" supposed to be "$0.7\mu$mol"? What am I missing?"

*Reply: Thank you for noticing it. It is supposed to be 0.7 molm$^{-2}$s$^{-1}$ and we have revised the text accordingly.*

4. "l. 143 – Explain this better, is system solving iteratively for leaf temperature that satisfies…. "meet the following physical flux equations:" Is that what this does?"

*Reply: Pardon for the confusion, we calculate the coupled photosynthesis and stomatal conductance system to find the value that satisfy Eq. (18) and (19), where we didn't iterate for leaf temperature. The intention in this text is to state the conditions that must be satisfied, while the*

*solution is defined in the following paragraph, Eq. (20) and Eq. (21). We will revise the text as follows: "Here, the leaf-level net carbon assimilation rate ($\overline{A}_{n,x}$), stomatal conductance for $CO_2$ ($G_{sc,x}$), and boundary layer conductance for $CO_2$ ($G_{bc}$) were calculated to satisfy the following physical flux equations".*

5. "Eq 21 looks strange. "$Ca-Rd$", Rd is a rate in umol/m2/s, but Ca is CO2 concentration. Does "$kp,xCa$" make it a rate too."

*Reply: Yes, you are correct, kp,xCa is also a rate. Pardon for the confusion, we have revised the unit in kp,x into molm$^{-2}$s$^{-1}$ for clarity.*

6. "l. 176-177 Very strange. How can you know "maximum Rubisco carboxylation rate at the canopy level ($Vcmax25,x$ ($l$))"? Strangely worded. Not really a whole canopy trait at all, because your reference is $Vcmax25(0)$. I would think that Vcmax is a characteristic of specific leaf N concentration maybe for upper leaves. OK, as you describe for $Vcmax25(0)$ on line 186."

*Reply: Pardon for the confusion, you are correct that Vcmax25,x(l) is characteristic for leaf. It is calculated from the maximum Rubisco carboxylation rate at a certain canopy depth, where depth is measured as the cumulative leaf area index (LAI) from the top of the canopy down to l. We have revised as follows: "$V_{cmax25,x}$ used in the photosynthesis module (Section 2.1) is obtained by dividing the maximum Rubisco carboxylation rate at a LAI depth of l ($V_{cmax25,x}(l)$) [...]".*

7. "l. 187-190 – This sentence implies conflict or difference, but in both cases $Vcmax25(0)$ is based on SLN for all three crops. Re-word to avoid that issue, or delete the whole sentence."

*Reply: Thank you for the clarification. You're correct. We have revised as follow: "Here, while Bonan et al. (2011) uses the fixed value of $V_{cmax25}(0)$ value over time, $V_{cmax25}(0)$ in MATCRO is calculated dynamically [...]"*

8. "Eq. 27 – I don't like having two equations for $Vcmax25(0)$ from two sources. That does not make sense."

*Reply: We agree that using two different sources may appear inconsistent and confusing. However, study from Bonelli and Fernando (2020) of Figure 1 shows the relationship between photosynthetic parameters and SLN varies considerably and decline photosynthetic activity is observed during the reproductive stage compared to the vegetative stage (Drouet and Bonhomme, 2004). We adopted stage specific parameterizations from this study to better capture this physiological difference since no single dataset adequately represents both growth phases. We will clarify this in the revised manuscript by adding this sentence: "Stage-specific parameterizations were applied to reflect the lower photosynthetic activity observed during the reproductive phase compared to the vegetative phase since no single dataset adequately represents both growth phases."*

9. "l. 220-235 and Figure 2 – Where are the equations and figures for partitioning to stem? Missing. Not in Table 1 either. At least mention and say "not shown", or is stem "by

difference". Also ear is not the same as grain. Tell us how you get to grain yield. Very approximately, grain is 85% of ear at maturity, but grain growth starts later than ear, actually a few days after flowering. So Kyld is about 0.85??? You use 0.83. OK"

*Reply: We thank the reviewer for this helpful comment. You are correct that we did not present the stem partition explicitly. In the model, partitioning to stem is represented as the remaining fraction after allocation to leaf and ear from the ratio to shoots/roots (line 256-257), and we will clarify this in the text.*

*For ear and grain, pardon us for the misleading statement. Ear is the storage organ which is not grain. We have parameterized Kyld from observation data and got the value of 0.83, hence we will use this value in the study. We will revise in the manuscript: "The term "ear" in maize represents the reproductive organ to store the grain. The grain developed later than the ear with approximately 83% of ear at maturity in this study (Table 1, kyld)."*

10. "Table 1 –SLW could be somewhat related to SLN. Please give Tb, $Th$, $To$ in Centigrade."

*Reply: We have revised Tb, Th, and To into Celcius*

11. "l. 260 – You call this validation. OK, if independent. But then, what data did you use for calibration? I suspect you used this data for calibration. Line 288-292 indicates that you calibrated life cycle to AgMIP data."

*Reply: We thank the reviewer for this important clarification. We will replace term "validation" to "evaluation" in order to avoid confusion. Calibration in this study was limited to phenological parameters (sowing, flowering, harvest) from AgMIP data (Table 2, Bassu et al., 2014) and biomass partitioning (leaf and ear) from Ciampitti et al. (2013 a,b). We then compare the simulation at the point-scale for phenology (flowering and maturity) and evaluated with LAI, aboveground biomass, and yield with AgMIP data (Table 2).*

12. "l. 278-279 – Confusing to go elsewhere for soil data, when you give the soil types of AgMIP study in the Table 2. Re-write."

*Reply:Thank you for noticing this. We will rewrite it into "We identified the soil texture from the gridded soil texture dataset of ISIMIP (Volkholz and Müller, 2020)".*

13. "l. 308-309 – You indicate N fertilization rates. What about N mineralization rates of each soil?"

*Reply: We thank the reviewer for this comment. In the current version of MATCRO-Maize, soil N mineralization is not explicitly simulated. Instead, its effect is represented implicitly through an empirical function of specific leaf nitrogen (SLN) and nitrogen fertilizer (Nfert) in Eq. (29) and Eq. (30). We acknowledge that this simplification limits the model's ability to capture the nitrogen dynamics in the soil, the limitation of this factor in the model will be written in the model limitation (the same as major concern point one).*

14. "l. 340-341 – I am confused. Here you reduced rubisco "rate" and SLN? On what basis? How was this justified (was it based on the validation data)? Apparently, you did calibrate to the data or thought about a possible reason."

*Reply: Thank you for raising this topic. We mentioned in the response to technical correction point 8, study of Bonelli and Fernando (2020) compared photosynthetic parameter varies across SLN from multiple studies. In MATCRO-Maize, we used the reduction in Rubisco activity based on experimental data of Drouet and Bonhomme (2004) with the intention to test the lower photosynthetic rate observed in this study as we have done in Figure 12. We understand writing this sentence in l. 340-341 may have caused confusion, hence we will move this sentence to the result section to explain about Figure 12.*

15. "Figure 6 for Brazil and others would indicate a problem with temperature parameterization for Vcmax(0), because you have a To that is too low, and even a Th is too low. You have values typical of a C-3 temperate warm-season crop."

*Reply: Wagree that site-specific varieties, such as those cultivated in Brazil may have higher optimum temperature and tolerances for photosynthesis than represented in our parameterization. In MATCRO-Maize, To and Th is not directly related to Vcmax but for phenological development. We used cardinal temperature for growing period as reported by Osborne et al. (2015) as a generalized representation of maize photosynthesis. While this approach does not capture potential variation in heat tolerance across regions or cultivars in site-scale, it provides a universal parameter for global-scale.*

16. "l.355-364 – These statistics and Figures 5, 6, and 7 indicate quite poor performance of MATCRO-Maize. Can we recommend a model that performs that poorly, for use by global gridded teams? Figure 7A and 7B use LAI and crop mass over time which is not warranted because of auto-correlation effects of time-series data (gives high correlation because it uses time-series values)."

*Reply:You are correct, we acknowledge Fig. 5, 6, and 7 indicate weak correlations. However, we will remove Fig. 7A and 7B (we also stated the same in reply for major concern point 3). In this study, Fig. 8, 10, and 11 demonstrate moderate and statistically significant correlations at country and global scales. While we recognize the limitations at site level, we consider MATCRO-Maize useful for global yield estimation, particularly in major producing regions, and will revise the text to highlight both strengths and limitations transparently.*

17. Figure 8 and 9 really seem to be "blind" evaluation because MATCRO is so much above the observed. Something is seriously missing here that causes the mis-match. Figure 8 shows MATCRO doing much better than warranted in drought-prone regions such as West Africa or Mexico or southwestern USA, so is the soil water balance failing or is stomatal conductance effect excessively conserving soil water? Or is it the "big-leaf" photosynthesis approach, very incomplete handling of N-fert effect on Vcmax, or something else? Figure 10 could point out issues with the soils for each country and stated N-fert that you used."

*Reply: We acknowledge that overestimations are evident in drought-prone regions as reviewer have mentioned or West and Southern Africa, central Brazil, and northern Argentina, where maize is mainly rainfed and exposed to drought with limited fertilizer inputs. This mismatch of the overestimation is likely linked to limitations in the soil water balance module, which may not fully capture soil variability and water stress. The soil water balance module in MATCRO perform poorly in representing the soil water balance in different soil depth. However, due to the limited availability of observational data on soil water dynamics, this explanation cannot be fully confirmed within the scope of this study. Other factors such as the big-leaf photosynthesis approach, the empirical treatment of nitrogen effects on Vcmax, and stomatal conductance responses, may also play a role. While these weaknesses highlight areas for future improvement, we believe the model remains informative for global-scale yield estimation.*

18. "Figure 11 (MATCRO usually over-estimates) differs from Figure 7 (where MATCRO under-estimates).  Any reasons for this?"

*Reply: Thank you for pointing them out. The apparent contradiction arises because Figure 7 shows site-level yield, it uses experimental conditions with no nitrogen fertilizer and no irrigation led to low yield.  While Figure 11 shows country-scale, yields are averaged across many grid cells where management inputs are different. The Brazil site in the experimental data shows underestimated value, while the country-scale shows overestimated value. Most maize grown in Brazil is rainfed, with some irrigated areas summarizes country-level averages across grid cells. Moreover, the universal parameterization can result in overestimation when applied under these heterogeneous conditions.*

19. "l. 414 and 417 – what do you mean by "changed parameters". Be more specific, is it what you mentioned on lines 340-341 without justification?"

*Reply: Thank you for pointing this out. We will replace the term "changed parameters" with "test of the SLN–Vcmax relationship" to avoid ambiguity.*

20. "Figure 13 – indicate source of N-fert values used for x-axis"

*Reply: We used Nfert values from gridded dataset (ISIMIP; Volkholz and Ostberg, 2022) and will put it in the revised manuscript.*

21. "l. 428 – replace "were statistically significant" with "showed statistically significant correlations" I also challenge "relatively well", as performance was not very good."

*Reply:Thank you for your suggestion, we agree with your point. We have replaced it in the revised manuscript.*

22. "l. 433 – "One reason" not "the reason"

*Reply: Thank you for your suggestion, we have adopted it in the revised manuscript.*

23. "l. 450 – Many maize models have LAI growth relatively uncoupled from photosynthesis and C balance. Carbon-driven LAI growth may cause problems."

*Reply: Thank you for the comment. We agree that driving LAI directly from carbon balance can create feedbacks that cause overestimation. To address this, should incorporate constraints on LAI development and leaf partitioning when LAI becomes unrealistically large. We will add this point explicitly in the limitations section: "A limitation of the current model is that LAI development is directly driven by carbon balance, which may create feedbacks leading to overestimation. Future improvements should incorporate constraints on LAI expansion and adjust leaf partitioning when LAI exceeds realistic levels."*

24. "Go back and confirm that is really how the Brazilian experiment was handled as $Nfert$ = 0. OR, this indicates that you have problems with getting soil N mineralization simulated. I did not see a word about SOC of soils."

*Reply: Pardon for the confusion. The Brazilian experiments relied on soil nitrogen mineralization rather than applied fertilizer, and this effect is implicitly represented through the SLN parameterization in MATCRO. We have revised sentences to clarify this point and explicitly note the lack of a mechanistic soil organic carbon and nitrogen mineralization module as a limitation.*

25. "L, 466 – and soil fertility"

*Reply: We will incorporate this into the manuscript, as soil fertility is also an important source of model error and contributes to spatial variation.*

26. "Table 4 – I am surprised that the other gridded global models for maize are performing that poorly. Correlation is a weak test."

*Reply: Thank you for raising this. We agree that correlation is a weak test at the grid scale. However, the moderate level of correlation is typical in global-scale evaluation for crop model due to the noise in yield data and uncertainty in management inputs. We will clarify this point in the manuscript.*

27. "l. 535-544 – Nfert problems suggest to me that MATCRO is very deficient in having, or totally lacking in a soil organic matter module and lacking in an semblance of a plant N balance. The authors need to come clean on this and say they lack a plant N balance module and lack a soil N module."

*Reply: We agree that MATCRO-Maize does not include a soil organic matter module or a plant nitrogen balance (as we also stated in major concern point 1). We mimic the relationship between Nfert and SLN from other studies (Muchow, 1988). The model does not simulate nitrogen cycling in soil or plants, and we will add this sentence: "Nitrogen effects are represented indirectly via SLN as a function of fertilizer rate and developmental stage, which constrains the model ability to capture nitrogen cycling in soils and plants."*

28. "l. 550 – replace "would be" with "are"

*Reply:We have adopted it in the manuscript.*

*References:*

- *Bassu, S., Brisson, N., Durand, J. L., Boote, K., Lizaso, J., Jones, J. W., Rosenzweig, C., Ruane, A. C., Adam, M., Baron, C., Basso, B., Biernath, C., Boogaard, H., Conijn, S., Corbeels, M., Deryng, D., De Sanctis, G., Gayler, S., Grassini, P., Hatfield, J., Hoek, S., Izaurralde, C., Jongschaap, R., Kemanian, A. R., Kersebaum, K. C., Kim, S. H., Kumar, N. S., Makowski, D., Müller, C., Nendel, C., Priesack, E., Pravia, M. V., Sau, F., Shcherbak, I., Tao, F., Teixeira, E., Timlin, D., and Waha, K.: How do various maize crop models vary in their responses to climate change factors?, Glob Chang Biol, 20, 2301–2320, https://doi.org/10.1111/gcb.12520, 2014.*
- *Ciampitti, I. A., Murrell, S. T., Camberato, J. J., Tuinstra, M., Xia, Y., Friedemann, P., and Vyn, T. J.: Physiological dynamics of maize nitrogen uptake and partitioning in response to plant density and N stress factors: I. Vegetative phase, Crop Sci, 53, 2105–2119, https://doi.org/10.2135/cropsci2013.01.0040, 2013a.*
- *Ciampitti, I. A., Murrell, S. T., Camberato, J. J., Tuinstra, M., Xia, Y., Friedemann, P., and Vyn, T. J.: Physiological dynamics of maize nitrogen uptake and partitioning in response to plant density and nitrogen stress factors: II. reproductive phase, Crop Sci, 53, 2588–2602, https://doi.org/10.2135/cropsci2013.01.0041, 2013b.*
- *Drouet, J. L. and Bonhomme, R.: Effect of 3D Nitrogen, Dry Mass per Area and Local Irradiance on Canopy Photosynthesis Within Leaves of Contrasted Heterogeneous Maize Crops, Ann Bot, 93, 699–710, https://doi.org/10.1093/aob/mch099, 2004.*
- *Muchow, R. C.: Effect of nitrogen supply on the comparative productivity of maize and sorghum in a semi-arid tropical environment I. Leaf growth and leaf nitrogen, Field Crops Res, 18, 1–16, https://doi.org/10.1016/0378-4290(88)90055-X, 1988.*
- *Osborne, T., Gornall, J., Hooker, J., Williams, K., Wiltshire, A., Betts, R., and Wheeler, T.: JULES-crop: a parametrisation of crops in the Joint UK Land Environment Simulator, Geosci. Model Dev, 8, 1139–1155, https://doi.org/10.5194/gmd-8-1139-2015, 2015.*
- *Volkholz, J. and Ostberg, S.: ISIMIP3a N-fertilizer input data (v1.2). ISIMIP Repository., https://doi.org/10.48364/ISIMIP.311496.2, 2022.*

---

## Author Comment (AC2)

*Dear RC2,*

*We sincerely appreciate the time and effort you dedicated to reviewing our work. Below, we provide our detailed responses to your concerns:*

**A. Model validation**

"1. It is recommended to use multiple gridded yield datasets to validate global gridded crop models (GGCMs) because grid-level yields can vary significantly between datasets, which is a significant source of uncertainty when assessing GGCM performance (Müller et al., 2017, Lin et al., 2021). Currently, annual gridded yield datasets are available for the globe and major producing countries at a 5-arcmin resolution (Su et al., 2022, Cao et al., 2025). In addition to comparing their model simulation with the Global Dataset of Historical Yields (GDHY), authors are encouraged to account for yield dataset uncertainty by comparing their model simulation with a family of recent gridded yield datasets."

*Reply: Thank you for this helpful suggestion. We agree that relying on a single yield dataset can lead to bias evaluation. Hence, we will add the comparison with other datasets as well for figure 8 and add the explanation "Simulated maize yields and three references in Figure 8 of MATCRO-Maize, Global Dataset of Historical Yield (GDHY by Iizumi and Sakai, 2019), GlobalCropYield (Cao et al., 2025), and Spatial Production Allocation Model (SPAM by IFRI, 2019) were harmonized into 0.5° resolution. The value was averaged over 1981–2014 for GDHY, averaged over 1982-2014 for GlobalCropYield, and year 2010 for SPAM.". In addition, we will include spatial distribution maps showing the differences between MATCRO-Maize yields and each reference dataset to make the comparison clearer in the revised manuscript.*

[Figure]

**Figure 8.** Global distribution of the 30-year average (1981-2010) maize yield by (a) simulations from the MATCRO-Maize and (b) the GDHY dataset. For comparison, yield estimates from shorter periods are also shown from (c) GlobalCropYield for 29-year average (1982-2014) and (d) SPAM2010 for year 2010. The simulated yield is aggregated based on the harvested area from MIRCA2000.

"2. The validation of crop phenology at the global level is currently lacking. I'm happy to see the model validation result at the site level (Fig. 4). However, the data compared are for only four sites and one year, which is inadequate for concluding the model performance. Gridded crop phenology datasets have recently become available for the globe and some major countries (Luo et al., 2020, Yang et al., 2020, Mori et al., 2023). I strongly encourage the authors to compare their simulation with these datasets."

*Reply: Thank you for insightful comments. In this study, we evaluated crop phenology at the point scale using a single growing-time dataset. As data availability is still limited for phenology, we focused on simulating the yield and relied on the gridded crop calendar from Jägermeyr et al. (2021) for planting and harvest dates. In response to your suggestion, we will expand the evaluation by comparing simulated phenology with recently available of global dataset of crop phenological events (GCPE by Mori et al., 2023) and present discrepancies in the figure for simulated harvest time compared with GGCMI and GCPE.*

"3. In relation to Comment#2, in the current form, it is unclear how the model parameter values related to crop phenology were determined before the model simulation. The authors state that "We used local daily climate data … and phenological data (planting, flowering, and maturity dates) for model input data at each site. (Line 276-278)". Did you calibrate the parameter values using the site data and then run the model? If so, this does not constitute model validation because no independent data were used for comparison. I would ask the authors to clarify this point and rerun the model validation if necessary."

*Reply: We are truly sorry for the confusion. You are correct, the current figures reflect the same data with parameterization which is not independent validation. We will use different term from "validation" and replace it with "evaluation". Figure 4 will be described as an evaluation of model fit for phenology (flowering and maturity dates) with caption: "Figure 4. Model-fit comparison of the flowering and maturity date simulations (SIM on the y-axis) and observations (OBS on the x-axis)."*

**B. Modeling**

"1. How did you determine Gdd,m (eq.22; the growing degree days at maturity)? Is this a universal value across grid cells worldwide? It is well-documented that Gdd,m varies spatially, with higher values in warmer regions and lower values in cooler regions (Deryng et al., 2011, Mori et al., 2023). I would ask the authors to clarify this."

*Reply:  Pardon us for the confusion. We use different value of growing degree days in each grid cell as noted in Deryng et al,. (2011) and Bouman et al., (2001). We will revise the Eq. 22 with adding the subscript of i for each grid cell where i means the grid cell number as stated below:*

$$D_{vs,i} = G_{dd,i}/G_{ddm,i},$$ (22)

"2. The leaf area index (LAI) simulated by the MATCRO-Maize model appeared to be lower than the site observations (Fig. 5). It is also noticed that the difference in maximum LAI between the

sites is smaller in the simulation than in the observations (Fig. 5). It leads to the thought that the maximum value of the specific leaf nitrogen parameterized with annual nitrogen application rate (*Nfert*) (eq. 29) is rather site-dependent and cannot be applied universally in its current form. This does not mean that publishing this preprint is unjustified. However, readers at least want to know whether underestimation of the seasonal maximum LAI correlates with environmental conditions, such as soil carbon content, soil total nitrogen content, water holding capacity of the soil and so on, in order to seek a possible scaling factor to convert specific leaf nitrogen to LAI. The equation (8) of Hasegawa et al. (2008) for the fraction of canopy cover may help the authors relate specific leaf nitrogen to seasonal maximum LAI (though this equation was developed for rice). If such a correlation analysis provides no insight, then calibrating the scaling factor for each country is another option, as was done by Ai and Hanasaki (2023)."

*Reply: Thank you for raising this topic. We agree that the simulated LAI is lower and shows less variation across sites compared to observations. In Figure 5, we applied the same universal parameters (SLW and leaf partitioning) across all sites, as our aim was global-scale application. Under low nitrogen conditions (e.g., Brazil), this can lead to underestimated LAI because the universal parameters do not represent no-fertilizer situations in the site scale simulation, as leaf morphological traits are known to vary with nitrogen availability (Ciampitti et al., 2013a,b; Hokmalipour and Darbandi, 2011). A sensitivity test in MATCRO confirmed that varying SLW strongly affects simulated LAI. The SLN–Vcmax relationship itself is applied globally because site-specific data are not available. Moreover, the soil water balance in MATCRO tends to underestimate water availability in deeper soil layers, which may contribute to yield underestimation under rainfed conditions. However, this could not be confirmed due to the limited availability of observational data. Other factors not considered in the current model framework may also contribute to this bias. We will clarify in the manuscript that the underestimation of LAI is more likely due to using universal morphological parameters at the site scale.*

"3. The presentation of the relationship between *Nfert* and yield, as presented in Fig. 12, is a bit misleading and could be improved. As can be seen in Fig. 12 (a), yield increases with an increase in *Nfert*, but then saturates. The yield response to *Nfert*, as derived from FAOSTAT, is consistent with literature which attributes recent maize yield growth to delayed leaf senescence (staygreen), morphological change from horizontal to vertical leaf type and increased drought tolerance, and resulting increase in planting density, rather than an increase in N input (Duvick, 2005). These genetics and management improvements have changed maize yield response to N input (Fig. 3 of DeBruin et al. 2017). Therefore, liner regression is inappropriate to describe the *Nfert*-yield relationship. Consider using a nonlinear regression or locally estimated scatterplot smoothing (LOWESS) instead. More importantly, the presented version of MATCRO-Maize imperfectly represent the *Nfert*-yield relationship (regardless of whether the data for Egypt is included or omitted). Rather than presenting Fig. 13, I would suggest the authors discuss this limitation of the model."

*Reply: Thank you very much for this constructive suggestion. We agree that the relationship between Nfert and yield cannot be adequately described by linear regression. We will use nonlinear regression (or LOWESS) in the revised manuscript and remove Fig. 13. We will also add a statement in the limitation section noting that MATCRO-Maize can generally reproduce*

*yield responses to different N fertilizer levels, but the model tends to overestimate yields under certain conditions (e.g., Egypt), which indicates opportunity for further improvement as follow:*

*"The current version of MATCRO-Maize can reproduce yield responses to nitrogen fertilization across a range of fertilizer levels, but it tends to overestimate yields under certain conditions (e.g., Egypt) likely because the model assumes higher nitrogen use efficiency and idealized irrigation conditions where actual yields are constrained by soil quality, management, and local cultivar traits that are not explicitly represented. This suggests that the representation of nitrogen effects in the model remains simplified, and further refinement is needed for region-specific scale simulation."*

"4. The simulated aboveground biomass was lower than the site observations (Fig. 7). However, the simulated yields at the country level were substantially overestimated. This discrepancy may be due to inaccurate partitioning to harvested organ or to stress factors reducing yield formation. I do understand that there are many factors not considered in the model, such as biotic stresses (pests and diseases, weeds, etc.), as described in Line 554. Nevertheless, recent crop models that are embedded in Earth system models that operate at a global level are encouraged to incorporate some form of parameterization to handle major drivers of historical yield growth even in a simple way (Lombardozzi et al., 2020). Alternatively, please consider calibrating some of the existing parameters to better reproduce historical yields (Ai and Hanasaki, 2023)."

*Reply: Thank you for this thoughtful comment and explanations. We acknowledge the discrepancy between site-level and country-level simulations due to the use of universal parameter in the site-level simulation. While additional calibration could improve agreement with historical yields, our approach emphasizes physiological mechanisms and universal parameters rather than statistical fitting. We will clarify this distinction in the manuscript and note the limitation that stress factors and other drivers of yield formation are not yet explicitly represented as follow:*

*"A limitation of the current study is the use of universal parameters at the site scale leads to discrepancies between site-level and country-level simulations. It partly arises from applying universal parameters across different environments. Moreover, genetic variation among cultivars is not considered, and key factors of yield formation are not yet explicitly represented (e.g. plant nitrogen balance)."*

**C. Technical corrections**

"1. 'Production' is generally measured in tones and is calculated by multiplying yield (production volume per unit harvested area and cropping season) by area harvested, in the case of single-season maize (see Box 1 of Wei et al., 2023). However, as MATCRO-Maize does not harvest area, the '"yield model" is more appropriate than the "production model".

"2. Line 264. I think the correct citation for the GDHY is "Iizumi and Sakai, 2020" rather than "Iizumi, 2019". Please check what recent literature describes this point (for instance, Data Availability and references of Iizumi et al., 2025)."

"3. Line 281. Do you mean "AgMERRA" (Ruane et al., 2015), a bias-corrected version of the MERRA reanalysis designed for agricultural applications, rather than the original "MERRA"?"

"4. Line 173. In agronomic literature, the flowering of maize is generally referred to as 'silking'. The first time it appears, you should mention this, for example, "flowering (known as silking; $Dvs,flw$)"."

*Reply: Thank you for clarification on the technical corrections. We agree with your review in point 1-4 and we will adopt them in the revised manuscript.*

*References:*

- *Bouman, B. A. M., Kropff, M., Tuong, T., Wopereis, M., ten Berge, H., and van Laar, H.: Oryza2000: modeling lowland rice, International Rice Research Institute, and Wageningen: Wageningen University and Research Centre, Philippines and Wageningen, the Netherlands, 235 pp., 2001.*
- *Ciampitti, I. A., Murrell, S. T., Camberato, J. J., Tuinstra, M., Xia, Y., Friedemann, P., and Vyn, T. J.: Physiological dynamics of maize nitrogen uptake and partitioning in response to plant density and N stress factors: I. Vegetative phase, Crop Sci, 53, 2105–2119, https://doi.org/10.2135/cropsci2013.01.0040, 2013a.*
- *Ciampitti, I. A., Murrell, S. T., Camberato, J. J., Tuinstra, M., Xia, Y., Friedemann, P., and Vyn, T. J.: Physiological dynamics of maize nitrogen uptake and partitioning in response to plant density and nitrogen stress factors: II. reproductive phase, Crop Sci, 53, 2588–2602, https://doi.org/10.2135/cropsci2013.01.0041, 2013b.*
- *Hokmalipour, S., Darbandi, M. H. Effects of nitrogen fertilizer on chlorophyll content and other leaf indicate in three cultivars of maize (Zea mays L.). World Applied Sciences Journal, 15(12), 1780-1785, 2011.*
- *International Food Policy Research Institute (IFPRI), Global Spatially-Disaggregated Crop Production Statistics Data for 2010 Version 2.0, https://doi.org/10.7910/DVN/PRFF8V, Harvard Dataverse, V4, 2019.*

- Ai, Z. and Hanasaki, N.: Simulation of crop yield using the global hydrological model H08 (crp.v1), Geosci. Model Dev., 16, 3275–3290, https://doi.org/10.5194/gmd-16-3275-2023, 2023.

- Cao, J., Zhang, Z., Luo, X. et al. Mapping global yields of four major crops at 5-minute resolution from 1982 to 2015 using multi-source data and machine learning. Sci Data 12, 357 (2025). https://doi.org/10.1038/s41597-025-04650-4

- DeBruin, J.L., Schussler, J.R., Mo, H. and Cooper, M. (2017), Grain Yield and Nitrogen Accumulation in Maize Hybrids Released during 1934 to 2013 in the US Midwest. Crop Science, 57: 1431-1446. https://doi.org/10.2135/cropsci2016.08.0704

- Deryng, D., W. J. Sacks, C. C. Barford, and N. Ramankutty (2011), Simulating the effects of climate and agricultural management practices on global crop yield, Global Biogeochem. Cycles, 25, GB2006, doi:10.1029/2009GB003765.

- Duvick, D. N. The contribution of breeding to yield advances in maize (Zea mays L.). Adv. Agron. 86, 83–145 (2005).

- Hasegawa, T., Sawano, S., Goto, S. et al. A model driven by crop water use and nitrogen supply for simulating changes in the regional yield of rain-fed lowland rice in Northeast Thailand. Paddy Water Environ 6, 73–82 (2008). https://doi.org/10.1007/s10333-007-0099-1

- Iizumi, T., Sakai, T. The global dataset of historical yields for major crops 1981–2016. Sci Data 7, 97 (2020). https://doi.org/10.1038/s41597-020-0433-7

- Toshichika Iizumi, Toru Sakai, Yoshimitsu Masaki, Kei Oyoshi, Takahiro Takimoto, Hideo Shiogama, Yukiko Imada, David Makowski, Assessing the capacity of agricultural research and development to increase the stability of global crop yields under climate change, PNAS Nexus, Volume 4, Issue 4, April 2025, pgaf099, https://doi.org/10.1093/pnasnexus/pgaf099

- Müller, C., Elliott, J., Chryssanthacopoulos, J., Arneth, A., Balkovic, J., Ciais, P., Deryng, D., Folberth, C., Glotter, M., Hoek, S., Iizumi, T., Izaurralde, R. C., Jones, C., Khabarov, N., Lawrence, P., Liu, W., Olin, S., Pugh, T. A. M., Ray, D. K., Reddy, A., Rosenzweig, C., Ruane, A. C., Sakurai, G., Schmid, E., Skalsky, R., Song, C. X., Wang, X., de Wit, A., and Yang, H.: Global gridded crop model evaluation: benchmarking, skills, deficiencies and implications, Geosci. Model Dev., 10, 1403–1422, https://doi.org/10.5194/gmd-10-1403-2017, 2017.

- Lin, T.-S., Song, Y., Lawrence, P., Kheshgi, H. S., & Jain, A. K. (2021). Worldwide maize and soybean yield response to environmental and management factors over the 20th and 21st centuries. Journal of Geophysical Research: Biogeosciences, 126, e2021JG006304. https://doi.org/10.1029/2021JG006304

- Lombardozzi, D. L., Lu, Y., Lawrence, P. J., Lawrence, D. M., Swenson, S., & Oleson, K. W., et al. (2020). Simulating agriculture in the Community Land Model Version 5. Journal of Geophysical Research: Biogeosciences, 125, e2019JG005529. https://doi.org/10.1029/2019JG005529

- Luo, Y., Zhang, Z., Chen, Y., Li, Z., and Tao, F.: ChinaCropPhen1km: a high-resolution crop phenological dataset for three staple crops in China during 2000–2015 based on leaf area index (LAI) products, Earth Syst. Sci. Data, 12, 197–214, https://doi.org/10.5194/essd-12-197-2020, 2020.

- Akira MORI, Yasuhiro DOI, Toshichika IIZUMI, GCPE: The global dataset of crop phenological events for agricultural and earth system modeling, Journal of Agricultural Meteorology, 2023, Volume 79, Issue 3, Pages 120-129, Released on J-STAGE July 10, 2023, Advance online publication May 16, 2023, Online ISSN 1881-0136, Print ISSN 0021-8588, https://doi.org/10.2480/agrmet.D-23-00004,

- Alex C. Ruane, Richard Goldberg, James Chryssanthacopoulos, 2015. Climate forcing datasets for agricultural modeling: Merged products for gap-filling and historical climate series estimation. Agricultural and Forest Meteorology, 200, 233-248. https://doi.org/10.1016/j.agrformet.2014.09.016

- Su, H., Willaarts, B., Luna-Gonzalez, D., Krol, M. S., and Hogeboom, R. J.: Gridded 5 arcmin datasets for simultaneously farm-size-specific and crop-specific harvested areas in 56 countries, Earth Syst. Sci. Data, 14, 4397–4418, https://doi.org/10.5194/essd-14-4397-2022, 2022.

- Yang Y, Ren W, Tao B et al., 2020: Characterizing spatiotemporal patterns of crop phenology across North America during 2000-2016 using satellite imagery and agricultural survey data. ISPRS Journal of Photogrammetry and Remote Sensing 170, 156-173. https://doi.org/10.1016/j.isprsjprs.2020.10.005

- Wei, D., Gephart, J.A., Iizumi, T. et al. Key role of planted and harvested area fluctuations in US crop production shocks. Nat Sustain 6, 1177–1185 (2023). https://doi.org/10.1038/s41893-023-01152-2

---

## Author Response (AR1)

**Dear Editor and Reviewers,**

We are pleased to resubmit our revised manuscript in response to the reviewers' valuable comments and suggestions. The feedback has been incredibly helpful in improving the quality of our work and we truly appreciate the time and effort you've dedicated to reviewing it.

**For RC1,**

**First, we will reply for your major concern:**

"1. The Nfert problems suggest to me that MATCRO is very deficient in having, or totally lacking, a soil organic matter module and lacking in plant N balance. The authors need to come clean on this and state in the text that MATCRO either has or lacks a plant N balance module and soil N module. Soils were adequately described for water balance processes, but there was no mention of soil organic matter or soil N mineralization. Basing SLN and Vcmax on Nfert is a very limiting approach and suggests that soil N supply is ignored. The authors need to state those limitations and improvements even more strongly."

Reply: We thank reviewer for this crucial comment and acknowledge that our earlier description did not sufficiently state this limitation. We understand that our model has limitations in fully capturing the nitrogen balance. However, the model includes a representation of leaf nitrogen content that covers a wide range of fertilizer levels from experimental data. This provides a practical approach to reflect variations in nitrogen availability across different conditions, implicitly accounting for soil N supply. We have clarified this limitation in Section 4.3 (L583-589) of the manuscript:

"MATCRO-Maize currently lacks explicit simulation of soil organic carbon and soil nitrogen mineralization. Instead, the effects of nitrogen supply are represented by describing the relationship between a broad range of nitrogen fertilization levels (Muchow, 1988) and specific leaf nitrogen (SLN), which subsequently affects photosynthetic capacity (Vcmax). While this simplification allows for global-scale application, it limits the model ability to represent nitrogen balance in maize yield at specific sites. Yield variations can be influenced by soil organic carbon and nitrogen, which are affected by farming practices and contribute to soil fertility (Ma et al., 2023). Future development could involve coupling MATCRO with a mechanistic soil nitrogen and carbon module to dynamic plant nitrogen balance. This would enhance the model ability to capture nitrogen dynamics under varying soil types and management practices.

"2. The authors use the term validation. Are Figures 5-10 independent data? But if so, then what data did you use for calibration? Was calibration only from literature for individual parameters? Need to indicate what data was used for calibration as that was never stated. Otherwise, readers will suspect you used this data for calibration."

Reply: Pardon us for the confusion, our manuscript did not clearly state calibration and evaluation. Figures 5-10 are based on independent data. Partitioning parameters for leaves, ear, and specific

leaf weight during the growing period were used for calibration by using biomass data derived from literature (Ciampitti et al., 2013a,b). The comparisons of LAI, biomass, and yield in Figures 5-7 for point scale, as well as yield in Figures 8-10 at the global-scale, were used only for evaluation and not for calibration.

We have replaced the term "validation" with "evaluation" to more accurately describe the process. Our approach involved evaluating the model's performance using independent data, without implying model verification or accuracy at the point-scale level (Figure 5-7). We also replaced the term "parameterized" with "calibrated".

To further clarify, we have added the following sentences:

- Clarifying parameters for calibration in L295 (Section 2.3.1): "Model calibration was conducted based on phenological data (Table 2, Bassu et al., 2014) and biomass data for carbon partitioning of leaf and ear derived from Ciampitti et al. (2013a, b). The model was assessed at the point scale to check the calibration for phenology (flowering and maturity) and was evaluated against time-series data of LAI, aboveground biomass, and harvested yield (see Section 3.1) that were not included in the model calibration."
- Clarifying model evaluation using independent data in discussion section of L367 (Section 3.1): "It is noted that Figures 5 7 present the model evaluation using independent data. The evaluation was performed using a global parameter from the literature to simulate plant organs in the global-scale simulation, which may have resulted in some deviations." .

  Moreover, we defined the term "global parameter" in Section 2.3.1 of L297: "In this study, a global parameter from the literature was applied uniformly across all regions at the grid-cell level instead of using site-specific calibrated parameters in the simulations.".
- "3. The statistics of Figures 5, 6, and 7 indicate quite poor performance of MATCRO-Maize and correlation statistics are weak tests. Can we recommend a model that performs that poorly for use by global gridded teams? Figure 7A and 7B are based on LAI and crop mass over time, but the high correlations there are misleading because of auto-correlation effects of time-series data (give high correlation because it uses time-series values)."

Reply: We agree that correlation based on time-series data lead to autocorrelation (e.g. LAI and biomass over time in Figs. 7A and 7B). To avoid misleading results, we have removed Figure 7A and 7B from the performance test (for LAI and total aboveground biomass).

Despite the poor model performance at the point scale, we are confident that MATCRO-Maize can provide valuable contribution to global gridded teams, because the model demonstrates a moderate correlation at the global scale (R = 0.54) and for the top 20 major producing countries (R = 0.58) when compared with FAO data (Figs 9 and 11, respectively). The poor model performance at the point scale is mainly attributed to the simulation method, where a global parameter set identified from the point scale datasets, i.e. "global parameters", were used. If we use parameters calibrated at each site, we could get high agreement with the observational data. To clarify this point, we added the following sentences:

L463 "The point-scale simulations were evaluated using global parameters to assess their ability to capture broad yield patterns across different regions. The simulated harvested yield were statistically significant at the point scale (Fig. 7), indicating that MATCRO-Maize model could simulate maize growth and yield, but its performance was limited at the point-scale. However, there were some discrepancies between the simulations and observations remain [...]. Global parameters at the point scale enable testing the model's applicability across various regions, although local variations in soil, climate, or crop management may not be fully captured."

**About your concern on technical correction:**

1. "1. 83 – Says C-3 here. Should be C-4."

Reply: You are correct, thank you for noticing this error. We have corrected the text accordingly in L81: "MATCRO-Maize calculates net carbon assimilation for the entire canopy (An) via the big-leaf model, where C4 leaf-level photosynthesis is separately calculated for sunlit and shaded leaves from the coupled photosynthesis—stomatal conductance model (Dai et al., 2004)."

2. "L 101 – why bother with "co-limited" photosynthesis? That is a C-3 hold-over and probably does not apply to C-4."

Reply: Pardon us for the confusion, Thank you for pointing this out. We understand that "colimited" photosynthesis mainly refers to C3 and is not appropriate for C4, hence we have replaced "co-limited photosynthesis" in the manuscript with "carbon fixation rate" in L99.

3. "1. 120 – Is " $0.7\mu$ " supposed to be " $0.7\mu$ mol"? What am I missing?"

Reply: Thank you for noticing it. It is supposed to be 0.7 molm-2s-1 and we have revised the text accordingly in L119 and Eq.(12).

4. "1. 143 – Explain this better, is system solving iteratively for leaf temperature that satisfies.... "meet the following physical flux equations:" Is that what this does?"

Reply: Pardon for the confusion, we calculate the coupled photosynthesis and stomatal conductance system to find the value that satisfy Eq. (18) and (19), where we didn't iterate for leaf temperature.

The intention in this text is to state the conditions that must be satisfied in Eq. (18) and (19), while the solution is defined in the following paragraph and Eq. (20) and Eq. (21). We will replace "meet" with "were calculated to satisfy" as follows in L140: "Here, the leaf-level net carbon assimilation rate  $(\overline{A}_{n,x})$ , stomatal conductance for  $CO_2(G_{sc,x})$ , and boundary layer conductance for  $CO_2(G_{bc})$  were calculated to satisfy the following physical flux equations".

5. "Eq 21 looks strange. "Ca-Rd", Rd is a rate in umol/m2/s, but Ca is CO2 concentration. Does "kp,xCa" make it a rate too."

Reply: Yes, you are correct, kp,xCa is also a rate. Pardon for the confusion, we have revised the unit in kp,x into molm-2s-1 for clarity in L108.

6. "l. 176-177 Very strange. How can you know "maximum Rubisco carboxylation rate at the canopy level (*Vcmax25,x* (*l*))"? Strangely worded. Not really a whole canopy trait at all, because your reference is *Vcmax25*(0). I would think that Vcmax is a characteristic of specific leaf N concentration maybe for upper leaves. OK, as you describe for *Vcmax25*(0) on line 186."

Reply: Pardon for the confusion, you are correct that Vcmax25,x(l) is characteristic for leaf. It is calculated from the maximum Rubisco carboxylation rate at a certain canopy depth, where depth is measured as the cumulative leaf area index (LAI) from the top of the canopy down to l. We have revised as follows in L175: " $V_{cmax25,x}$  used in the photosynthesis module (Section 2.1) is obtained by dividing the maximum Rubisco carboxylation rate at a LAI depth of l ( $V_{cmax25,x}(l)$ ) f...]".

7. "1. 187-190 – This sentence implies conflict or difference, but in both cases *Vcmax*25(0) is based on SLN for all three crops. Re-word to avoid that issue, or delete the whole sentence."

Reply: Thank you for the clarification. We have deleted "in each plant functional type" and revised as follow in L185: "Here, while Bonan et al. (2011) uses the fixed value of  $V_{cmax25}(0)$  value over time,  $V_{cmax25}(0)$  in MATCRO is calculated dynamically [...]"

8. "Eq. 27 – I don't like having two equations for *Vcmax*25(0) from two sources. That does not make sense."

Reply: We agree that using two different sources may appear inconsistent and confusing. However, study from Bonelli and Fernando (2020) of Figure 1 shows the relationship between photosynthetic parameters and SLN varies considerably and decline photosynthetic activity is observed during the reproductive stage compared to the vegetative stage (Drouet and Bonhomme, 2004).

We adopted stage specific parameterizations from this study to better capture this physiological difference since no single dataset adequately represents both growth phases. We will clarify this in the revised manuscript by adding this sentence in L196: "Stage-specific parameterizations were applied to reflect the lower photosynthetic activity observed during the reproductive phase compared to the vegetative phase since no single dataset adequately represents both growth phases."

9. "1. 220-235 and Figure 2 – Where are the equations and figures for partitioning to stem? Missing. Not in Table 1 either. At least mention and say "not shown", or is stem "by difference". Also ear is not the same as grain. Tell us how you get to grain yield. Very approximately, grain is 85% of ear at maturity, but grain growth starts later than ear, actually a few days after flowering. So Kyld is about 0.85??? You use 0.83. OK"

Reply: We thank the reviewer for this helpful comment. You are correct that we did not present the stem partition explicitly. In the model, partitioning to stem is represented as the remaining fraction after allocation to leaf and ear from the ratio to shoots/roots and we clarified this in L228: "The stem partitioning was determined by reducing the shoot ratio with respect to the leaf and ear"

About how to calculate ear, we apologize for the misleading statement. The ear is a storage organ, we agree that it is not the grain itself. We have derived Kyld from experimental data with a value of 0.83, which was used in the study. We revised in the manuscript in L223-225: "The term "ear" in maize represents the organ that supports the development and storage of grain. The grain developed later than the ear with approximately 83% of ear at maturity in this study (see Section 2.2.5).

10. "Table 1 –SLW could be somewhat related to SLN. Please give Tb, Th, To in Centigrade."

Reply: We have revised the unit of Tb,Th,and To into Celcius in Table 1.

11. "l. 260 – You call this validation. OK, if independent. But then, what data did you use for calibration? I suspect you used this data for calibration. Line 288-292 indicates that you calibrated life cycle to AgMIP data."

Reply: We thank the reviewer for this important clarification. You are correct that this data is not independent. In order to avoid confusion, we have replaced the term "validation" with "evaluation" as we stated in the *major concern point two* where we have replaced "parameterized" and "calibrated" in the sections of Phenology (2.2.1, L170), Crop Growth (2.2.3, L227), and Model evaluation at a point scale (Section 2.3.1, L290) to clearly indicate which data were used for calibration. We also replaced Figure 4 from "Comparison" into "Model-fit".

12. "1. 278-279 – Confusing to go elsewhere for soil data, when you give the soil types of AgMIP study in the Table 2. Re-write."

Reply: We have revised the sentence (L281) into "We identified the soil texture from the gridded soil texture dataset of ISIMIP (Volkholz and Müller, 2020)".

13. "1. 308-309 – You indicate N fertilization rates. What about N mineralization rates of each soil?"

Reply: In the current version of MATCRO-Maize, soil N mineralization is not explicitly simulated. Instead, its effect is represented implicitly through an empirical function of specific leaf nitrogen (SLN) and nitrogen fertilizer (Nfert) in Eq. (29) and Eq. (30). We acknowledge that this simplification limits the model's ability to capture the nitrogen dynamics in the soil, the limitation of this factor in the model will be written in the model limitation (already included in the reply of *major concern point one*).

14. "l. 340-341 – I am confused. Here you reduced rubisco "rate" and SLN? On what basis? How was this justified (was it based on the validation data)? Apparently, you did calibrate to the data or thought about a possible reason."

Reply: Thank you for raising this topic. A study of Bonelli and Fernando (2020) compared photosynthetic parameter varies across SLN from multiple studies. In MATCRO-Maize, we used the reduction in Rubisco activity based on experimental data of Drouet and Bonhomme (2004) with the intention to test the lower photosynthetic rate observed in this study as we have done in Figure 13 (refer to the revised manuscript). We understand writing this sentence in Section 2.3.2 may have caused confusion, hence we have moved this sentence into new Section of 3.3 (The effects of photosynthesis and N fertilizer) to explain about Figure 13.

We have added this sentences in L444: "Figure 13 illustrates the comparison of country-level yield data with nitrogen fertilizer levels: (a) FAOSTAT data, (b) simulated yield by MATCRO-Maize, (c) the impact of reduced Rubisco activity on photosynthetic rates based on experimental data from Drouet and Bonhomme (2004) in the "test Sln-Vcmax" scenario, and (d) the effect of reduced photosynthetic rates due to lower initial specific leaf nitrogen at planting time in the "test Sln,plt" scenario."

15. "Figure 6 for Brazil and others would indicate a problem with temperature parameterization for Vcmax(0), because you have a To that is too low, and even a Th is too low. You have values typical of a C-3 temperate warm-season crop."

Reply: We agree that site-specific varieties, such as those cultivated in Brazil may have higher optimum temperature and tolerances for photosynthesis than represented in our parameterization. In MATCRO-Maize, To and Th is not directly related to Vcmax but for phenological development. We used cardinal temperature for growing period reported by Osborne et al. (2015) as a generalized representation of maize photosynthesis. While this approach does not capture potential variation in heat tolerance across regions or cultivars in site-scale, it provides a universal parameter for global-scale.

16. "1.355-364 – These statistics and Figures 5, 6, and 7 indicate quite poor performance of MATCRO-Maize. Can we recommend a model that performs that poorly, for use by global gridded teams? Figure 7A and 7B use LAI and crop mass over time which is not warranted because of auto-correlation effects of time-series data (gives high correlation because it uses time-series values)."

Reply: You are correct, we acknowledge Fig. 5, 6, and 7 indicate weak correlations, despite being statistically significant. However, we removed Fig. 7A and 7B (we stated in *major concern point* 3). In this study, Fig. 10, 11, and 12 (refer to revised manuscript) demonstrate moderate and statistically significant correlations at country and global scales. While we recognize the limitations at site level, we consider MATCRO-Maize useful for global yield estimation, particularly in major producing regions.

We added the limitation and strength in L605: "Although MATCRO-Maize shows relatively weak correlations at the site scale due to the use of generalized parameters that do not account for local varieties and management, the model demonstrates consistent and statistically significant performance at country and global levels. This indicates that MATCRO-Maize is well suited for capturing large-scale yield patterns and for application in global gridded crop modeling, while recognizing its limited capacity for precise site-specific prediction."

17. Figure 8 and 9 really seem to be "blind" evaluation because MATCRO is so much above the observed. Something is seriously missing here that causes the mis-match. Figure 8 shows MATCRO doing much better than warranted in drought-prone regions such as West Africa or Mexico or southwestern USA, so is the soil water balance failing or is stomatal conductance effect excessively conserving soil water? Or is it the "big-leaf" photosynthesis approach, very incomplete handling of N-fert effect on Vcmax, or something else? Figure 10 could point out issues with the soils for each country and stated N-fert that you used."

Reply: We acknowledge that overestimations are evident in drought-prone regions as reviewer have mentioned or West and Southern Africa, central Brazil, and northern Argentina, where maize is mainly rainfed and exposed to drought with limited fertilizer inputs. This mismatch of the overestimation is likely linked to limitations in the soil water balance module, which may not fully capture soil variability and water stress. The soil water balance module in MATCRO perform poorly in representing the soil water balance in deeper soil depth. However, due to the limited availability of observational data on soil water dynamics, this explanation cannot be fully confirmed within the scope of this study.

Other factors such as the big-leaf photosynthesis approach, the empirical treatment of nitrogen effects on Vcmax, and stomatal conductance responses, may also play a role. While these weaknesses highlight areas for future improvement, we believe the model remains informative for global-scale yield estimation. Future improvements, particularly in refining the soil water balance and incorporating more detailed processes for nitrogen effects and stomatal conductance, will enhance the model accuracy for local-scale application.

18. "Figure 11 (MATCRO usually over-estimates) differs from Figure 7 (where MATCRO under-estimates). Any reasons for this?"

Reply: Thank you for pointing out this contradiction. The difference between Figure 7 and Figure 11 arises from the different scales and conditions represented. Figure 7 shows site-level yield under experimental conditions with no nitrogen fertilizer and no irrigation, leading to low yields. In contrast, Figure 11 shows country-scale yield, where the data are averaged across multiple grid cells with varying management practices, including fertilizer use and irrigation, which can lead to overestimation. Specifically, in Brazil, the experimental site underestimates yield due to limited inputs, while the country-scale data, which includes both rainfed and irrigated areas, results in overestimated yield. The use of a universal (global) parameterization can also contribute to overestimation in regions with heterogeneous conditions like Brazil.

19. "1. 414 and 417 – what do you mean by "changed parameters". Be more specific, is it what you mentioned on lines 340-341 without justification?"

Reply: Thank you for pointing this out. We replaced the term "changed parameters" with "reduced effect of photosynthesis" in L451 to avoid ambiguity.

20. "Figure 13 – indicate source of N-fert values used for x-axis"

Reply: We have added explanation that Nfert values from gridded dataset (ISIMIP; Volkholz and Ostberg, 2022) for explaining Figure 13 (refer to revised manuscript) in L447: "The nitrogen fertilizer values were derived from gridded dataset (ISIMIP; Volkholz and Ostberg, 2022).".

21. "l. 428 – replace "were statistically significant" with "showed statistically significant correlations" I also challenge "relatively well", as performance was not very good."

Reply: Thank you for your suggestion, we agree with your point. We have replaced it in L464: "The simulated harvested yield showed statistically significant correlations at the point scale (Fig. 7), indicating that the MATCRO-Maize model could simulate maize growth and yield, but its performance was limited at the point-scale."

```
22. "1. 433 – "One reason" not "the reason"
```

Reply: Thank you for your suggestion, we have adopted it in the revised manuscript in L475: "One potential factor contributing to the underestimation of the".

23. "1. 450 – Many maize models have LAI growth relatively uncoupled from photosynthesis and C balance. Carbon-driven LAI growth may cause problems."

Reply: Thank you for the comment. We agree that driving LAI directly from carbon balance can create feedbacks that cause overestimation. To address this, we should incorporate constraints on LAI development and leaf partitioning when LAI becomes unrealistically large.

We have added L609: "However, global-scale simulation results tend to overestimate yield due to LAI being directly driven by carbon balance, which can create feedbacks that produce excessively high LAI. Future improvements should incorporate constraints on LAI expansion and adjust leaf partitioning when LAI exceeds realistic levels."

24. "Go back and confirm that is really how the Brazilian experiment was handled as *Nfert* = 0. OR, this indicates that you have problems with getting soil N mineralization simulated. I did not see a word about SOC of soils."

Reply: The Brazilian experiments relied on soil nitrogen mineralization rather than applied fertilizer. This effect is implicitly represented through the SLN parameterization in MATCRO, when nitrogen fertilizer is set to 0, there is still nitrogen in the leaf (Figure 1 in the manuscript).

We have revised sentences to clarify this point and explicitly note the lack of a mechanistic soil organic carbon and nitrogen mineralization module as a limitation in L586: "Yield variations can be influenced by soil organic carbon and nitrogen, which are affected by farming practices and contribute to soil fertility (Ma et al., 2023)."

```
25. "L, 466 – and soil fertility"
```

Reply: We have incorporated this into the manuscript, as soil fertility is also an important source of model error and contributes to spatial variation. It is included in previous reply (point 24).

26. "Table 4 – I am surprised that the other gridded global models for maize are performing that poorly. Correlation is a weak test."

Reply: We agree that correlation is a relatively weak metric at the grid scale. However, moderate correlations are typical in global-scale evaluations of crop models due to inherent noise in yield data and uncertainty in management inputs across regions.

We added this sentence to clarify that the observed level of correlation is expected and consistent with global-scale model evaluation practices in L544 in Section 4.2: "The moderate correlations observed reflect the typical influence of yield data variability and uncertainty in management practices across regions.".

27. "1. 535-544 – Nfert problems suggest to me that MATCRO is very deficient in having, or totally lacking in a soil organic matter module and lacking in an semblance of a plant N balance. The authors need to come clean on this and say they lack a plant N balance module and lack a soil N module."

Reply: We agree that MATCRO-Maize does not include a soil organic matter module or a plant nitrogen balance. We mimic the relationship between Nfert and SLN from other studies (Muchow, 1988). The model does not simulate nitrogen cycling in soil or plants, and we added this sentence to clarify the limitation of the model in L599: "Nitrogen effects are represented indirectly via SLN as a function of fertilizer rate and developmental stage, which constrains the model ability to capture nitrogen cycling in soils and plants."

28. "1. 550 – replace "would be" with "are"

Reply: We have adopted it in the manuscript in L615.

**References:**

- Bassu, S., Brisson, N., Durand, J. L., Boote, K., Lizaso, J., Jones, J. W., Rosenzweig, C., Ruane, A. C., Adam, M., Baron, C., Basso, B., Biernath, C., Boogaard, H., Conijn, S., Corbeels, M., Deryng, D., De Sanctis, G., Gayler, S., Grassini, P., Hatfield, J., Hoek, S., Izaurralde, C., Jongschaap, R., Kemanian, A. R., Kersebaum, K. C., Kim, S. H., Kumar, N. S., Makowski, D., Müller, C., Nendel, C., Priesack, E., Pravia, M. V., Sau, F., Shcherbak, I., Tao, F., Teixeira, E., Timlin, D., and Waha, K.: How do various maize crop models vary in their responses to climate change factors?, Glob Chang Biol, 20, 2301–2320, https://doi.org/10.1111/gcb.12520, 2014.
- Ciampitti, I. A., Murrell, S. T., Camberato, J. J., Tuinstra, M., Xia, Y., Friedemann, P., and Vyn, T. J.: Physiological dynamics of maize nitrogen uptake and partitioning in response to plant density and N stress factors: I. Vegetative phase, Crop Sci, 53, 2105–2119, https://doi.org/10.2135/cropsci2013.01.0040, 2013a.
- Ciampitti, I. A., Murrell, S. T., Camberato, J. J., Tuinstra, M., Xia, Y., Friedemann, P., and Vyn, T. J.: Physiological dynamics of maize nitrogen uptake and partitioning in response to plant density and nitrogen stress factors: II. reproductive phase, Crop Sci, 53, 2588–2602, https://doi.org/10.2135/cropsci2013.01.0041, 2013b.
- Drouet, J. L. and Bonhomme, R.: Effect of 3D Nitrogen, Dry Mass per Area and Local Irradiance on Canopy Photosynthesis Within Leaves of Contrasted Heterogeneous Maize Crops, Ann Bot, 93, 699–710, https://doi.org/10.1093/aob/mch099, 2004.
- Ma, Y., Dominic W., Mingsheng F., Lei Q., Rong L., and Johannes L. Global crop production increase by soil organic carbon. Nature Geoscience 16 (12): 1159-1165, 2023.

- Muchow, R. C.: Effect of nitrogen supply on the comparative productivity of maize and sorghum in a semiarid tropical environment I. Leaf growth and leaf nitrogen, Field Crops Res, 18, 1–16, https://doi.org/10.1016/0378-4290(88)90055-X, 1988.
- Osborne, T., Gornall, J., Hooker, J., Williams, K., Wiltshire, A., Betts, R., and Wheeler, T.: JULES-crop: a parametrisation of crops in the Joint UK Land Environment Simulator, Geosci. Model Dev, 8, 1139–1155, https://doi.org/10.5194/gmd-8-1139-2015, 2015.
- Volkholz, J. and Ostberg, S.: ISIMIP3a N-fertilizer input data (v1.2). ISIMIP Repository., https://doi.org/10.48364/ISIMIP.311496.2, 2022.
* * *
Next page

**For RC2,**

First, we will reply for your concerns based on three parts (model validation, modeling method, and technical correction)

**A. Model validation**

"1. It is recommended to use multiple gridded yield datasets to validate global gridded crop models (GGCMs) because grid-level yields can vary significantly between datasets, which is a significant source of uncertainty when assessing GGCM performance (Müller et al., 2017, Lin et al., 2021). Currently, annual gridded yield datasets are available for the globe and major producing countries at a 5-arcmin resolution (Su et al., 2022, Cao et al., 2025). In addition to comparing their model simulation with the Global Dataset of Historical Yields (GDHY), authors are encouraged to account for yield dataset uncertainty by comparing their model simulation with a family of recent gridded yield datasets."

Reply: Thank you for this helpful suggestion. We agree that relying on a single yield dataset can lead to bias evaluation. We compared the simulated yields with the dataset you mentioned and observed smaller variations in 5-arcmin resolution (Cao et al., 2025; IFPRI, 2019).

We have added Fig. 9 and explanation as follows:

**Figure 9(manuscript).** Global distribution of the 30-year average (1981-2010) maize yield by (a) simulations from the MATCRO-Maize and (b) the GDHY dataset. For comparison, yield estimates from shorter periods are also shown from (c) GlobalCropYield for 29-year average (1982-2014) and (d) SPAM2010 for year 2010. The simulated yield is aggregated based on the harvested area from MIRCA2000.

L403: "All datasets were harmonized to a 0.5° resolution, including simulated yield from MATCRO-Maize (Fig. 8(a)), the Global Dataset of Historical Yield (GDHY, Iizumi and Sakai, 2020; Fig. 8(b)), GlobalCropYield (GCY, f et al., 2025; Fig. 8(c)), and the Spatial Production Allocation Model by (SPAM, IFPRI, 2019; Fig. 8(d)). The data were averaged over 1981–2014 for GDHY, 1982–2014 for GlobalCropYield, and for the year 2010 for SPAM. While the overestimation could be seen mainly in tropical regions, the simulated yield could capture high-yielding regions, including the Corn Belt in the United States and the northern part of China, in agreement with the reference datasets."

"2. The validation of crop phenology at the global level is currently lacking. I'm happy to see the model validation result at the site level (Fig. 4). However, the data compared are for only four sites and one year, which is inadequate for concluding the model performance. Gridded crop phenology datasets have recently become available for the globe and some major countries (Luo et al., 2020, Yang et al., 2020, Mori et al., 2023). I strongly encourage the authors to compare their simulation with these datasets."

Reply: Thank you for your comment. Based on your recommendation, we have also compared our simulations with other available gridded phenology datasets in global-scale simulation as shown in Figure 8 for harvest time. We have added Fig. 8 and explanations as follow:

Figure 8. The difference between simulated harvest time (days) in MATCRO-Maize simulations with (a) GGCMI in the rainfed, and (b) irrigated conditions, (c) GCPE in the irrigated, and (d) rainfed conditions. Blue indicates

underestimation, while red indicates overestimation between simulation and references. Panels (e) and (f) show the mean of absolute differences (days) under the rainfed (a, c) and irrigated (b, d) comparisons, respectively.

L386: "The timing of seasonal biological events (i.e. harvest time) has a significant impact on crop growth and yield outcomes. Global yield is affected by global phenology. We assessed agreement by comparing the difference between simulated global harvest time (1981–2010 mean) with gridded global dataset of phenological datasets of GGCMI (Jägermeyr et al., 2021; Figs. 8 (a) and (b)), and GCPE (Mori et al., 2023; Figs. 8 (c) and (d)). The maps show consistent spatial patterns for later harvest time between the simulation and the reference datasets, in parts of Brazil, USA, southern and central Africa. The discrepancies between dataset are likely produced due to the difference in phenology parameterization and management assumptions where GGCMI and GCPE used different methodology and data sources. Moreover, the use of the growing degree day method in the simulations led to year-to-year differences in harvest time compared with the reference crop calendar used for the input data (Figs. 8 (a) and (b)). The mean absolute differences in harvest time (Figs. 8 (e) and (f)) indicated that the largest biases occur mostly in tropical regions."

Moreover, we defined the method in L324: "MATCRO-Maize was assessed for the phenological simulation of harvest time against the phenological dataset GGCMI (Jägermeyr et al., 2021) and global datasets of crop phenological events for agricultural and earth system modeling which was derived from various field experiments and a phenology model (GCPE; Mori et al., 2023). These datasets were compared under both rainfed and irrigated conditions in  $0.5^{\circ} \times 0.5^{\circ}$  resolution to check the model performance.".

"3. In relation to Comment#2, in the current form, it is unclear how the model parameter values related to crop phenology were determined before the model simulation. The authors state that "We used local daily climate data ... and phenological data (planting, flowering, and maturity dates) for model input data at each site. (Line 276-278)". Did you calibrate the parameter values using the site data and then run the model? If so, this does not constitute model validation because no independent data were used for comparison. I would ask the authors to clarify this point and rerun the model validation if necessary."

Reply: We calibrated the phenology parameters before running the model. Figure 4 is not the model validation, but it is the assessment to check the calibrated model. To clarify this point, we added a sentence and revised the caption as follows:

L298: "The model was then assessed at the point scale to check the calibration for phenology (flowering and maturity) and evaluated against time-series data of LAI, aboveground biomass, and harvested yield (see Section 3.1) that were not included in the model calibration."

Figure 4: "Model-fit comparison of the flowering and maturity date simulations (SIM on the y-axis) and observations (OBS on the x-axis"

**B. Modeling**

"1. How did you determine Gdd,m (eq.22; the growing degree days at maturity)? Is this a universal value across grid cells worldwide? It is well-documented that Gdd,m varies spatially, with higher

values in warmer regions and lower values in cooler regions (Deryng et al., 2011, Mori et al., 2023). I would ask the authors to clarify this."

Reply: Pardon us for the confusion. We use different value of growing degree days in each grid cell as noted in Deryng et al,. (2011) and Bouman et al., (2001). We have revised the Eq. 22 with adding the subscript of i for each grid cell where i means the grid cell number as stated below:

$$^{\prime\prime}D_{vs,i} = G_{dd,i}/G_{ddm,i}, \tag{22}$$

where  $G_{dd,i}$  is the growing degree days at t (time) for specific grid cell i,  $G_{ddm,i}$  is the growing degree day at maturity, [...]"

"2. The leaf area index (LAI) simulated by the MATCRO-Maize model appeared to be lower than the site observations (Fig. 5). It is also noticed that the difference in maximum LAI between the sites is smaller in the simulation than in the observations (Fig. 5). It leads to the thought that the maximum value of the specific leaf nitrogen parameterized with annual nitrogen application rate (*Nfert*) (eq. 29) is rather site-dependent and cannot be applied universally in its current form. This does not mean that publishing this preprint is unjustified. However, readers at least want to know whether underestimation of the seasonal maximum LAI correlates with environmental conditions, such as soil carbon content, soil total nitrogen content, water holding capacity of the soil and so on, in order to seek a possible scaling factor to convert specific leaf nitrogen to LAI. The equation (8) of Hasegawa et al. (2008) for the fraction of canopy cover may help the authors relate specific leaf nitrogen to seasonal maximum LAI (though this equation was developed for rice). If such a correlation analysis provides no insight, then calibrating the scaling factor for each country is another option, as was done by Ai and Hanasaki (2023)."

Reply: Thank you for raising this topic. We agree that the simulated LAI is lower and shows less variation across sites compared to observations. This can mainly be attributed to the use of global parameters, as you pointed out. In Figure 5, we applied the same parameters universally (specific leaf weight (SLW) and leaf partitioning) across all sites, as our aim was global-scale application. Under low nitrogen conditions (e.g., Brazil), it led to underestimated LAI because the universal parameters do not represent no-fertilizer situations in the site scale simulation, as leaf morphological traits are known to vary with nitrogen availability (Ciampitti et al., 2013a,b; Hokmalipour and Darbandi, 2011). We used higher SLW parameter in simulating point-scale leaf biomass in Brazil which led to underestimation. A sensitivity test in MATCRO confirmed that varying high SLW strongly lower simulated LAI. The SLN–Vcmax relationship itself is applied globally because site-specific data are not available.

Additionally, the soil water balance in MATCRO tends to underestimate water availability in deeper soil layers, which may further contribute to yield underestimation under rainfed conditions, though this could not be fully confirmed due to limited observational data. We also agree that the consideration of the canopy cover fraction (Hasegawa et al., 2008) could improve our model performance.

We have added this sentence to clarify the limitation in L468-471: "The underestimation of LAI is primarily due to the use of universal morphological parameters at the site scale. Future work will

improve site-specific performance by coupling LAI to key soil properties (soil organic carbon, total nitrogen, and water-holding capacity) and by incorporating canopy cover fraction following Hasegawa et al. (2008)".

"3. The presentation of the relationship between *Nfert* and yield, as presented in Fig. 12, is a bit misleading and could be improved. As can be seen in Fig. 12 (a), yield increases with an increase in *Nfert*, but then saturates. The yield response to *Nfert*, as derived from FAOSTAT, is consistent with literature which attributes recent maize yield growth to delayed leaf senescence (staygreen), morphological change from horizontal to vertical leaf type and increased drought tolerance, and resulting increase in planting density, rather than an increase in N input (Duvick, 2005). These genetics and management improvements have changed maize yield response to N input (Fig. 3 of DeBruin et al. 2017). Therefore, liner regression is inappropriate to describe the *Nfert*-yield relationship. Consider using a nonlinear regression or locally estimated scatterplot smoothing (LOWESS) instead. More importantly, the presented version of MATCRO-Maize imperfectly represent the *Nfert*-yield relationship (regardless of whether the data for Egypt is included or omitted). Rather than presenting Fig. 13, I would suggest the authors discuss this limitation of the model."

Reply: Thank you very much for this constructive suggestion. We agree that the relationship between Nfert and yield cannot be adequately described by linear regression. We have used polynomial regression (refers to Fig. 13 in the revised manuscript) and removed the corresponding graph without Egypt at Fig. 13. To clarify this point, we have modified Fig. 13 as follows:

Figure 13.  $N_{fert}$  impact on yield of (a) FAOSTAT, (b) simulated yield with the original setting (Default), (c) simulated yield with the changed  $S_{ln}$ - $V_{cmax}$  relationship (test Sln-Vcmax), (d) simulated yield with the changed parameter related to the  $D_{vs}$ - $S_{ln}$  function (test Sln, plt).  $N_{fert}$  (N fertilizer) and country yield were averaged across 30 years for each country. The legends for symbols are the same as those in Fig. 11. The solid lines are fitted curve for the data, while the dashed lines in (b), (c), and (d) indicate fitted curve based on the data in (a). All lines were fitted using a quadratic polynomial regression.

L449: "Figures 13 (a) and (b) show the comparisons based on Nfert for each FAOSTAT and simulated yield, respectively. MATCRO has a strong Nfert effect on the yield reflected in the steep upward trend of the fitted curves. This effect was scarcely alleviated by the intentionally reduced effect of photosynthesis (Figs. 13(c and d)), mainly because of the effect of Egypt as an outlier with higher values. Without Egypt as an outlier, the curves for FAOSTAT and MATCRO-Maize were more comparable. The maize yield in Egypt shows high value compared to other countries where significant overestimation was observed."

Model limitation L526 in Section 4.2: "The current version of MATCRO-Maize can reproduce yield responses to nitrogen fertilization across a range of fertilizer levels, but it tends to overestimate yields under certain conditions (e.g., Egypt) likely because the model assumes higher nitrogen use efficiency and idealized irrigation conditions where actual yields are constrained by soil quality, management, and local cultivar traits that are not explicitly represented. This suggests that the representation of nitrogen effects in the model remains simplified, and further refinement is needed for region-specific scale simulation.".

"4. The simulated aboveground biomass was lower than the site observations (Fig. 7). However, the simulated yields at the country level were substantially overestimated. This discrepancy may be due to inaccurate partitioning to harvested organ or to stress factors reducing yield formation. I do understand that there are many factors not considered in the model, such as biotic stresses (pests and diseases, weeds, etc.), as described in Line 554. Nevertheless, recent crop models that are embedded in Earth system models that operate at a global level are encouraged to incorporate some form of parameterization to handle major drivers of historical yield growth even in a simple way (Lombardozzi et al., 2020). Alternatively, please consider calibrating some of the existing parameters to better reproduce historical yields (Ai and Hanasaki, 2023)."

Reply: Thank you for this thoughtful comment and explanations. We acknowledge the discrepancy between site-level and country-level simulations due to the use of universal parameter in the site-level simulation. While additional calibration could improve agreement with historical yields, our approach emphasizes physiological mechanisms and universal parameters rather than statistical fitting.

We will clarify this distinction in the manuscript and note the limitation that stress factors and other drivers of yield formation are not yet explicitly represented as follow in L603 in Section 4.3:

"A limitation of the current study is the use of universal parameters at the site scale leads to discrepancies between site-level and country-level simulations. It partly arises from applying global parameters across different environments. Although MATCRO-Maize shows relatively weak correlations at the site scale due to the use of generalized parameters that do not account for local varieties and management, the model demonstrates consistent and statistically significant performance at country and global levels. This indicates that MATCRO-Maize is well suited for capturing large-scale yield patterns and for application in global gridded crop modeling, while recognizing its limited capacity for precise site-specific prediction."

**C. Technical corrections**

"1. 'Production' is generally measured in tones and is calculated by multiplying yield (production volume per unit harvested area and cropping season) by area harvested, in the case of single-season maize (see Box 1 of Wei et al., 2023). However, as MATCRO-Maize does not harvest area, the "yield model" is more appropriate than the "production model".

Reply: We agree with the term you have mentioned. We have revised the title from "production model" to "yield model".

"2. Line 264. I think the correct citation for the GDHY is "Iizumi and Sakai, 2020" rather than "Iizumi, 2019". Please check what recent literature describes this point (for instance, Data Availability and references of Iizumi et al., 2025)."

Reply: Thank you for clarification. We have checked and we revised the citation as "Iizumi and Sakai, 2020) in the manuscript accordingly.

"3. Line 281. Do you mean "AgMERRA" (Ruane et al., 2015), a bias-corrected version of the MERRA reanalysis designed for agricultural applications, rather than the original "MERRA"?"

Reply: You are correct, we used the bias-corrected version for Agriculture. Pardon us for missing this point. We have revised it in L283 along with the citation.

"4. Line 173. In agronomic literature, the flowering of maize is generally referred to as 'silking'. The first time it appears, you should mention this, for example, "flowering (known as silking; *Dvs,flw*)"."

Reply: Thank you for clarification on the term. We have revised it in L171.

**References:**

- Bouman, B. A. M., Kropff, M., Tuong, T., Wopereis, M., ten Berge, H., and van Laar, H.: Oryza2000: modeling lowland rice, International Rice Research Institute, and Wageningen: Wageningen University and Research Centre, Philippines and Wageningen, the Netherlands, 235 pp., 2001.
- Ciampitti, I. A., Murrell, S. T., Camberato, J. J., Tuinstra, M., Xia, Y., Friedemann, P., and Vyn, T. J.: Physiological dynamics of maize nitrogen uptake and partitioning in response to plant density and N stress factors: I. Vegetative phase, Crop Sci, 53, 2105–2119, https://doi.org/10.2135/cropsci2013.01.0040, 2013a.
- Ciampitti, I. A., Murrell, S. T., Camberato, J. J., Tuinstra, M., Xia, Y., Friedemann, P., and Vyn, T. J.: Physiological dynamics of maize nitrogen uptake and partitioning in response to plant density and nitrogen stress factors: II. reproductive phase, Crop Sci, 53, 2588–2602, https://doi.org/10.2135/cropsci2013.01.0041, 2013b.
- Hokmalipour, S., Darbandi, M. H. Effects of nitrogen fertilizer on chlorophyll content and other leaf indicate in three cultivars of maize (Zea mays L.). World Applied Sciences Journal, 15(12), 1780-1785, 2011.
- International Food Policy Research Institute (IFPRI), Global Spatially-Disaggregated Crop Production Statistics Data for 2010 Version 2.0, https://doi.org/10.7910/DVN/PRFF8V, Harvard Dataverse, V4, 2019.
- Ai, Z. and Hanasaki, N.: Simulation of crop yield using the global hydrological model H08 (crp.v1), Geosci. Model Dev., 16, 3275–3290, https://doi.org/10.5194/gmd-16-3275-2023, 2023.
- Cao, J., Zhao Z., Xiangzhong L., Yuchuan L., Jialu X., Jun X., Jichong H., and Fulu T. Mapping global yields of four major crops at 5-minute resolution from 1982 to 2015 using multi-source data and machine learning. Scientific Data 12, 1, 357, 2025.

- DeBruin, J.L., Schussler, J.R., Mo, H. and Cooper, M. (2017), Grain Yield and Nitrogen Accumulation in Maize Hybrids Released during 1934 to 2013 in the US Midwest. Crop Science, 57: 1431-1446. https://doi.org/10.2135/cropsci2016.08.0704
- Deryng, D., W. J. Sacks, C. C. Barford, and N. Ramankutty (2011), Simulating the effects of climate and agricultural management practices on global crop yield, Global Biogeochem. Cycles, 25, GB2006, doi:10.1029/2009GB003765.
- Duvick, D. N. The contribution of breeding to yield advances in maize (Zea mays L.). Adv. Agron. 86, 83–145 (2005).
- Hasegawa, T., Sawano, S., Goto, S. et al. A model driven by crop water use and nitrogen supply for simulating changes in the regional yield of rain-fed lowland rice in Northeast Thailand. Paddy Water Environ 6, 73–82 (2008). https://doi.org/10.1007/s10333-007-0099-1
- Iizumi, T., Sakai, T. The global dataset of historical yields for major crops 1981–2016. Sci Data 7, 97 (2020). https://doi.org/10.1038/s41597-020-0433-7
- Toshichika Iizumi, Toru Sakai, Yoshimitsu Masaki, Kei Oyoshi, Takahiro Takimoto, Hideo Shiogama, Yukiko Imada, David Makowski, Assessing the capacity of agricultural research and development to increase the stability of global crop yields under climate change, PNAS Nexus, Volume 4, Issue 4, April 2025, pgaf099, https://doi.org/10.1093/pnasnexus/pgaf099
- Müller, C., Elliott, J., Chryssanthacopoulos, J., Arneth, A., Balkovic, J., Ciais, P., Deryng, D., Folberth, C., Glotter, M., Hoek, S., Iizumi, T., Izaurralde, R. C., Jones, C., Khabarov, N., Lawrence, P., Liu, W., Olin, S., Pugh, T. A. M., Ray, D. K., Reddy, A., Rosenzweig, C., Ruane, A. C., Sakurai, G., Schmid, E., Skalsky, R., Song, C. X., Wang, X., de Wit, A., and Yang, H.: Global gridded crop model evaluation: benchmarking, skills, deficiencies and implications, Geosci. Model Dev., 10, 1403–1422, https://doi.org/10.5194/gmd-10-1403-2017, 2017.
- Lin, T.-S., Song, Y., Lawrence, P., Kheshgi, H. S., & Jain, A. K. (2021). Worldwide maize and soybean yield response to environmental and management factors over the 20th and 21st centuries. Journal of Geophysical Research: Biogeosciences, 126, e2021JG006304. https://doi.org/10.1029/2021JG006304
- Lombardozzi, D. L., Lu, Y., Lawrence, P. J., Lawrence, D. M., Swenson, S., & Oleson, K. W., et al. (2020). Simulating agriculture in the Community Land Model Version 5. Journal of Geophysical Research: Biogeosciences, 125, e2019JG005529. https://doi.org/10.1029/2019JG005529
- Luo, Y., Zhang, Z., Chen, Y., Li, Z., and Tao, F.: ChinaCropPhen1km: a high-resolution crop phenological dataset for three staple crops in China during 2000–2015 based on leaf area index (LAI) products, Earth Syst. Sci. Data, 12, 197–214, https://doi.org/10.5194/essd-12-197-2020, 2020.
- Mori A., Doi, Y., Iizumi, T. GCPE: The global dataset of crop phenological events for agricultural and earth system modeling, Journal of Agricultural Meteorology, 79, 3, 120-129, https://doi.org/10.2480/agrmet.D-23-00004, 2023.
- Alex C. Ruane, Richard Goldberg, James Chryssanthacopoulos, 2015. Climate forcing datasets for agricultural modeling: Merged products for gap-filling and historical climate series estimation. Agricultural and Forest Meteorology, 200, 233-248. https://doi.org/10.1016/j.agrformet.2014.09.016
- Su, H., Willaarts, B., Luna-Gonzalez, D., Krol, M. S., and Hogeboom, R. J.: Gridded 5 arcmin datasets for simultaneously farm-size-specific and crop-specific harvested areas in 56 countries, Earth Syst. Sci. Data, 14, 4397– 4418, https://doi.org/10.5194/essd-14-4397-2022, 2022.
- Yang Y, Ren W, Tao B et al., 2020: Characterizing spatiotemporal patterns of crop phenology across North America during 2000-2016 using satellite imagery and agricultural survey data. ISPRS Journal of Photogrammetry and Remote Sensing 170, 156-173. https://doi.org/10.1016/j.isprsjprs.2020.10.005
- Wei, D., Gephart, J.A., Iizumi, T. et al. Key role of planted and harvested area fluctuations in US crop production shocks. Nat Sustain 6, 1177–1185 (2023). https://doi.org/10.1038/s41893-023-01152-2

---

## Author Response (AR2)

**Dear Editor and Reviewers.**

"I appreciate the authors' effort in revising the manuscript. My comments have been satisfactorily addressed. The following are only suggestions for minor edits. I would be happy to suggest the acceptance of the manuscript for publication. 1. L24-25. "One of the reasons for this overestimation could be related to the strong nitrogen fertilization effect observed in MATCRO-Maize." Please consider rephrasing this sentence to avoid the confusion that this states about the CO2 fertilization effect. One suggested adjustment is "One of the reasons for this overestimation could be related to the strong model response to nitrogen fertilizer observed in MATCRO-Maize." 2. L251. "the calculation of evaporation" This would be evapotranspiration rather than evaporation. 3. L336. Typo. "290year period"."

Reply: We are pleased to hear the news. Thank you very much for the thoughtful efforts that reviewers and editors have taken in this manuscript. Hereby, we have revised as follows:

- 1. Thank you for your comment. We have replaced the related sentence with your recommended wording in **L23**: "One of the reasons for this overestimation could be related to the strong model response to nitrogen fertilizer observed in MATCRO-Maize".
- 2. We have replaced "evaporation" with evapotranspiration in **L252**, which the related process also includes the transpiration.
- 3. Thank you for pointing it out. We have revised the typographical error in **L342**: "[...], 29-year period [...]"

We have also revised our short summary to exclude abbreviations (See Note below) and rotated Table 4 to portrait orientation, as requested by the editorial team in the system notification.

Additionally, we revised certain sentences in the manuscript to enhance clarity and readability in the PDF file with the track changes.

| Sincerely,               |
|--------------------------|
| Astrid Yusara            |
| On behalf of all authors |
|                          |

Note: Short Summary (<500 characters)

We developed a maize version of a process-based crop model coupled to a land-surface model by incorporating photosynthesis for C4 plants and maize-specific parameters. The model was calibrated with field data and literature, and it was extensively validated with global reference yields. The model effectively captured interannual yield variability in global and county-level yield data, demonstrating its potential for assessing the climate impacts on maize production.

\_\_\_\_\_